# The ribosomal protein Asc1/RACK1 is required for efficient translation of short mRNAs

**Mary K Thompson, Maria F Rojas-Duran, Paritosh Gangaramani, Wendy V Gilbert***

Department of Biology, Massachusetts Institute of Technology, Cambridge, United States

**Abstract** Translation is a core cellular process carried out by a highly conserved macromolecular machine, the ribosome. There has been remarkable evolutionary adaptation of this machine through the addition of eukaryote-specific ribosomal proteins whose individual effects on ribosome function are largely unknown. Here we show that eukaryote-specific Asc1/RACK1 is required for efficient translation of mRNAs with short open reading frames that show greater than average translational efficiency in diverse eukaryotes. *ASC1* mutants in *S. cerevisiae* display compromised translation of specific functional groups, including cytoplasmic and mitochondrial ribosomal proteins, and display cellular phenotypes consistent with their gene-specific translation defects. Asc1-sensitive mRNAs are preferentially associated with the translational 'closed loop' complex comprised of eIF4E, eIF4G, and Pab1, and depletion of eIF4G mimics the translational defects of *ASC1* mutants. Together our results reveal a role for Asc1/RACK1 in a length-dependent initiation mechanism optimized for efficient translation of genes with important housekeeping functions.

**\*For correspondence:** wgilbert@mit.edu

**Competing interests:** The authors declare that no competing interests exist.

## Introduction

Ribosomes are universal protein-synthesizing machines that are highly conserved in their structure and function throughout all kingdoms of life. However, each domain of life has evolved unique ribosomal proteins that are added to the conserved core. The fundamental tasks of ribosomes — deciphering the genetic code and synthesizing peptide bonds — are the same in all organisms, so the functions of these 'extra' ribosomal proteins are intriguing, yet almost entirely unknown.

Eukaryotic ribosomes contain 13 domain-specific proteins that may play roles in translation initiation, which is both more complicated and more highly regulated in eukaryotes than in prokaryotes (*Ban et al., 2014*; *Sonenberg and Hinnebusch, 2009*). Recruitment of prokaryotic ribosomes to mRNAs requires only three initiation factors, IF1, 2, and 3, and relies on base-pairing between the RNA of the small ribosomal subunit and the anti-Shine-Delgarno sequence of the mRNA (*Boelens and Gualerzi, 2002*). In contrast, translation initiation in eukaryotes requires at least 12 initiation factors and proceeds by a complex series of steps beginning with recognition of the mRNA 5' cap structure, followed by unwinding of mRNA secondary structure, recruitment of the small (40S) ribosomal subunit, scanning, recognition of the initiation codon, and finally, joining of the large (60S) ribosomal subunit to form a functional ribosome (*Aitken and Lorsch, 2012*). Although the eukaryotic ribosome is generally considered to be a passive player during canonical initiation, several of its proteins have been implicated in mRNA recruitment. For example, RPL38 is required for the translation of the Hox body-patterning genes during embryonic development, allowing spatiotemporal regulation of gene expression through translational control (*Kondrashov et al., 2011*). Other proteins including RPS25, RPL40, and RACK1 are essential for the translation of viral mRNAs that are

**eLife digest** Ribosomes are structures within cells that are responsible for making proteins. Molecules called messenger RNAs (or mRNAs), which contain genetic information derived from the DNA of a gene, pass through ribosomes that then "translate" that information to build proteins. Although all living cells contain ribosomes, the protein building blocks that make up the structure of the ribosome are not the same in all species. Furthermore, the exact roles that each building block plays during translation are not known.

The ribosomes of plants, animals, and budding yeast contain the same protein, known as Asc1 in budding yeast and RACK1 in plants and animals. Thompson et al. have now explored the role of Asc1 in yeast cells by measuring translation in the absence of Asc1 using a technique called ribosome footprint profiling. This analysis revealed that cells lacking Asc1 translate fewer short mRNA molecules than normal cells. Short mRNAs encode small proteins that tend to play important 'housekeeping' roles in the cell — by forming the structural building blocks of ribosomes, for example.

It has been observed previously that short mRNAs are translated at a higher rate than longer mRNAs on average, although the reasons behind this bias are still mysterious. The findings of Thompson et al. suggest that the ribosome itself may discriminate between short and long mRNAs and that the Asc1 protein is involved in calibrating the ribosome's preference for short mRNAs.

Cells need differing amounts of small proteins in different growth conditions. It will therefore be interesting to investigate whether mRNA length discrimination can be regulated by Asc1 and/or other components of the ribosome to tune gene expression to the environment.

recruited to the ribosome via alternative initiation pathways (*Cherry et al., 2005*; *Landry et al., 2009*; *Lee et al., 2013*; *Majzoub et al., 2014*).

The eukaryote-specific ribosomal protein RACK1 is a WD40-repeat β-propeller protein that binds the solvent-exposed face of the 40S subunit near the mRNA exit channel, in close proximity to proteins that contact the mRNA during translation initiation (*Pisarev et al., 2008*; *Sengupta et al., 2004*). In addition to its function as a core ribosomal protein, in mammalian cells, RACK1 has been found in complex with several proteins involved in signal transduction including protein kinase C, Src kinase, and cAMP phosphodiesterase, among many others (*Adams et al., 2011*). The location of RACK1 on the ribosome together with its interactions with signaling proteins suggests a possible role in conveying stimulus-dependent information to the translation machinery (*Nilsson et al., 2004*). However, signaling pathways in yeast and human have diverged significantly compared to genes required for ribosomal function (*Stuart et al., 2003*), suggesting that RACK1 might have another, more conserved function during translation.

Loss of RACK1 causes widespread and pleiotropic defects in many organisms. Deletion of the *RACK1* homolog in budding yeast, *ASC1*, leads to slow growth, loss of invasive growth, loss of cell wall integrity, and decreased 60S subunit levels, among many described effects (*Li et al., 2009*; *Melamed et al., 2010*; *Valerius et al., 2007*; *Yoshikawa et al., 2011*). In metazoans, RACK1 is required for cell migration, neural tube closure, and control of post-synaptic excitation in the brain (*Kiely et al., 2009*; *Ron et al., 1999*; *Wehner et al., 2011*; *Yaka et al., 2002*). These cellular functions may explain why homozygous *RACK1* loss-of-function mutations cause early developmental lethality in mouse and flies (*Kadrmas et al., 2007*; *Volta et al., 2013*). However, it is not known whether and how the effects of RACK1 on ribosome function contribute to the myriad cellular and organismal phenotypes observed in *RACK1/ASC1* mutants (*Gibson, 2012*).

Here we have examined the translational functions of Asc1/RACK1 genome-wide by ribosome footprint profiling in yeast *ASC1* mutants. We show that Asc1 is required for the efficient translation of short mRNAs, including those encoding cytoplasmic and mitochondrial ribosomal proteins. This requirement is specific as deletion of other ribosomal proteins does not cause similar translation defects. Using translational reporters we demonstrate that length per se determines translational sensitivity to Asc1, thus confirming a role for Asc1 in the translational privileging of short mRNAs, which is a dominant trend in genome-wide translational efficiency data from diverse eukaryotes.

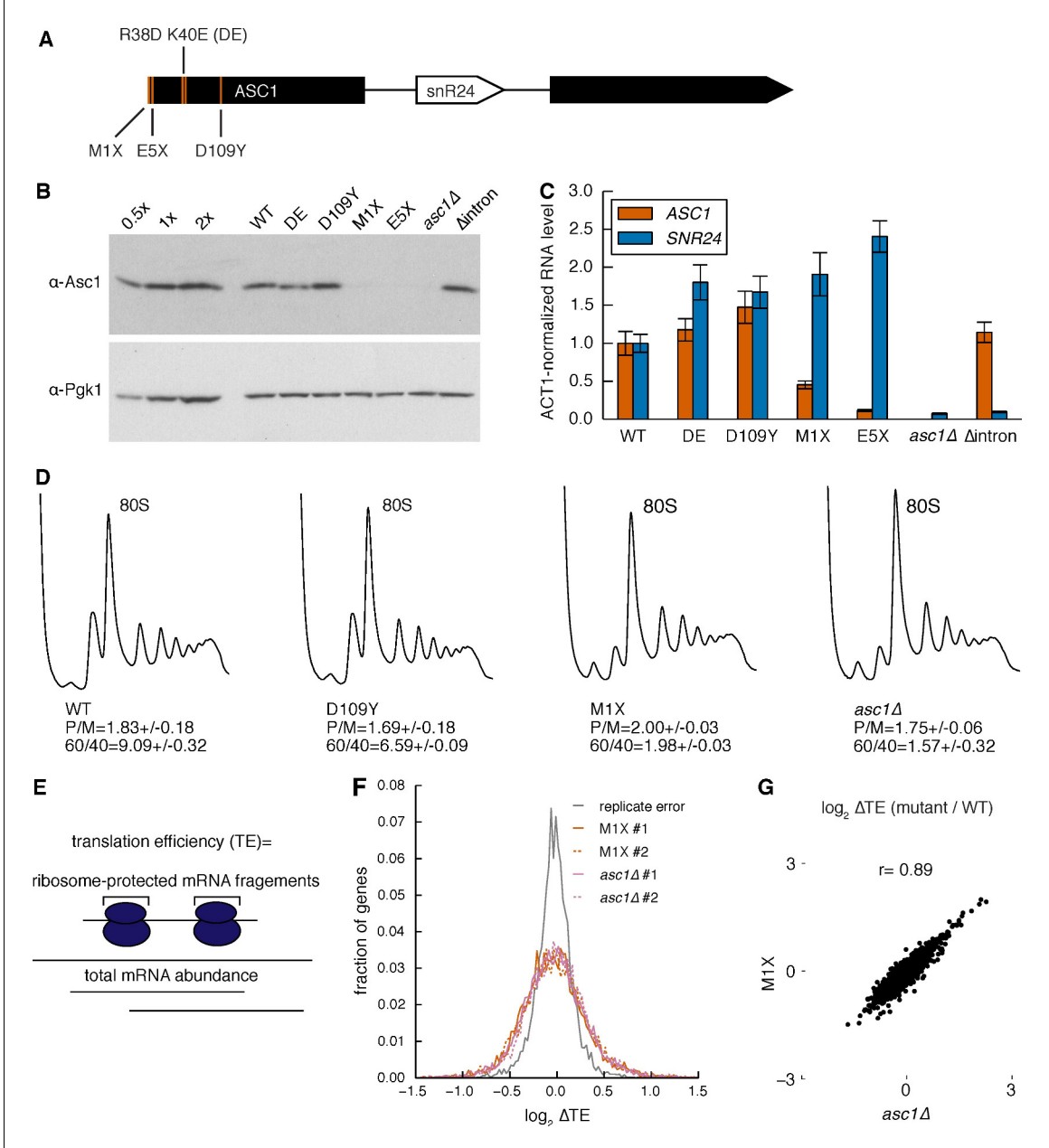

**Figure 1.** Loss of the Asc1 protein causes widespread changes in translation efficiency. (**A**) Gene model of *ASC1*, showing the *SNR24* snoRNA and location of protein null (M1X and E5X) and ribosome binding (DE and D109Y) mutations. (**B**) Asc1 protein levels quantified by Western blot. Pgk1 blot on the same membrane is shown as a loading control. Dilutions of the WT sample are shown on the left. Data is representative of three biological replicates. (**C**) *ASC1* mRNA and *SNR24* snoRNA levels quantified by qRT-PCR. Levels were normalized to *ACT1* mRNA levels. Error bars represent s.d. from three technical replicates. Data is representative of three biological replicates. (**D**) Polysome profiles of the *ASC1* mutants at 30°C. The polysome/monosome (P/M) ratio and 60S/40S (60/40) ratio are shown with s.d. from two biological replicates. (**E**) Calculation of translation efficiency as the ratio of ribosome-protected mRNA fragments to total mRNA abundance. (**F**) Distribution of changes in TE comparing two biological replicates from WT cells (i.e. replicate error) or *asc1*-M1X or *asc1*Δ to its corresponding WT comparison. #1 and #2 denote biological replicate experiments. (**G**) Scatterplot of TE changes between the two *ASC1* null mutants. The Pearson correlation coefficient is shown.

The following figure supplement is available for figure 1:

**Figure supplement 1.** The M1X mutation rescues the temperature-sensitive ribosome biogenesis defect of *asc1*Δ.

Remarkably, mRNA enrichment with proteins that mediate the formation of a 'closed loop' during translation — eIF4E, eIF4G, and Pab1 — is strongly biased towards short mRNAs and predicts Asc1-sensitivity, suggesting a role for Asc1 in closed-loop-dependent ribosome recruitment. Consistent with this prediction, we find that depletion of the central closed loop factor eIF4G mimics the translational effects of mutating *ASC1*. Finally, we show that loss of *ASC1* reduces mitochondrial translation and renders cells unable to use alternative carbon sources that require enhanced mitochondrial function, demonstrating the functional significance of translational perturbation in *ASC1* mutants. Together, our results reveal a role for Asc1 in the enhanced translation of short mRNAs and establish a direct connection between gene-specific effects of Asc1 on translation and defects in cellular physiology. Furthermore, because mitochondria are essential for energy generation and regulation of many cellular networks, our results suggest that the pleiotropic phenotypes associated with the Asc1/RACK1 protein should be re-examined in the context of mitochondrial health.

## Results

### Loss of the Asc1 protein perturbs global translation

The *ASC1* locus encodes two distinct gene products — the Asc1 protein and an intronic small nucleolar RNA, snR24. Because snR24 directs 2′-O-methylation of 25S rRNA at positions C1437, C1449, and C1450, some of the reported phenotypes of *ASC1* null mutants (*asc1Δ*) could be due to effects of deleting *SNR24* on ribosome biogenesis or function. In addition, Asc1/RACK1 may have functions off the ribosome (*Baum et al., 2004*; *Coyle et al., 2009*; *Warner and McIntosh, 2009*). We therefore created an allelic series of yeast mutants with altered Asc1 function to enable direct comparison of cellular and translational effects of Asc1/RACK1 (*Figure 1A*). We created protein null alleles by mutating a codon early in the *ASC1* ORF to a stop codon (*asc1-M1X* and *asc1-E5X*, where X denotes a stop codon), which abolished Asc1 protein expression but maintained wild type levels of *SNR24* (*Figure 1B,C*). Although bulk polysomes appeared normal in these strains, both *asc1Δ* and *asc1-M1X* showed reduced levels of free 60S subunits (*Figure 1D*). This slight discrepancy between our results and the literature (*Li et al., 2009*) may stem from differences in strain backgrounds because the Sigma1278b strain used here has higher free 60S subunit levels than S288C. Restoring *SNR24* expression rescued the temperature-sensitive polysome defect of the *asc1Δ* mutant in agreement with previous observations (*Figure 1—figure supplement 1A–D*) (*Li et al., 2009*). Both *asc1-M1X* and *asc1Δ* grow slowly under standard laboratory conditions, whereas a mutant lacking only snR24 grows as well as wild type, further demonstrating the importance of the Asc1 protein (*Figure 1—figure supplement 1E*).

To define the translational function of Asc1, we subjected the *ASC1* mutants to ribosome footprint profiling and RNA-seq. Together, these techniques allow quantification of ribosome densities transcriptome-wide and can be used to infer changes in gene-specific translation activity (*Ingolia et al., 2009*). Loss of the Asc1 protein caused changes in translation activity for many mRNAs as measured by translational efficiency (TE) — the number of ribosomal footprints normalized by the number of total RNA fragments for each mRNA (*Figure 1E–G*). The magnitude and pervasiveness of translation changes in *asc1-M1X* and *asc1Δ* are notable given the normal appearance of bulk polysomes in *ASC1* mutants (*Figure 1D*). Thus superficially normal polysomes can conceal significant perturbations of cellular translation. Together, these results demonstrate that the lack of Asc1 substantially alters the translational landscape of yeast cells.

### *ASC1* 'ribosome-binding' mutants associate with ribosomes and are largely functional

Next we examined isogenic yeast strains that express normal levels of Asc1 protein with perturbed association to the ribosome. Asc1 is a WD-repeat protein that interacts with helices 39 and 40 of the 18S rRNA primarily through its N-terminal blade (*Sengupta et al., 2004*). Directed mutation of several basic residues in this region interferes with the ribosome-binding capacity of the protein, with the strongest defect observed in the R38D K40E (DE) mutant (*Coyle et al., 2009*; *Sengupta et al., 2004*). Another Asc1 ribosome-binding mutant, D109Y, was discovered serendipitously in a forward genetic screen for mutants with defects in no-go decay, a ribosome-associated RNA quality control mechanism (*Kuroha et al., 2010*). These mutant proteins were expressed at near wild type levels

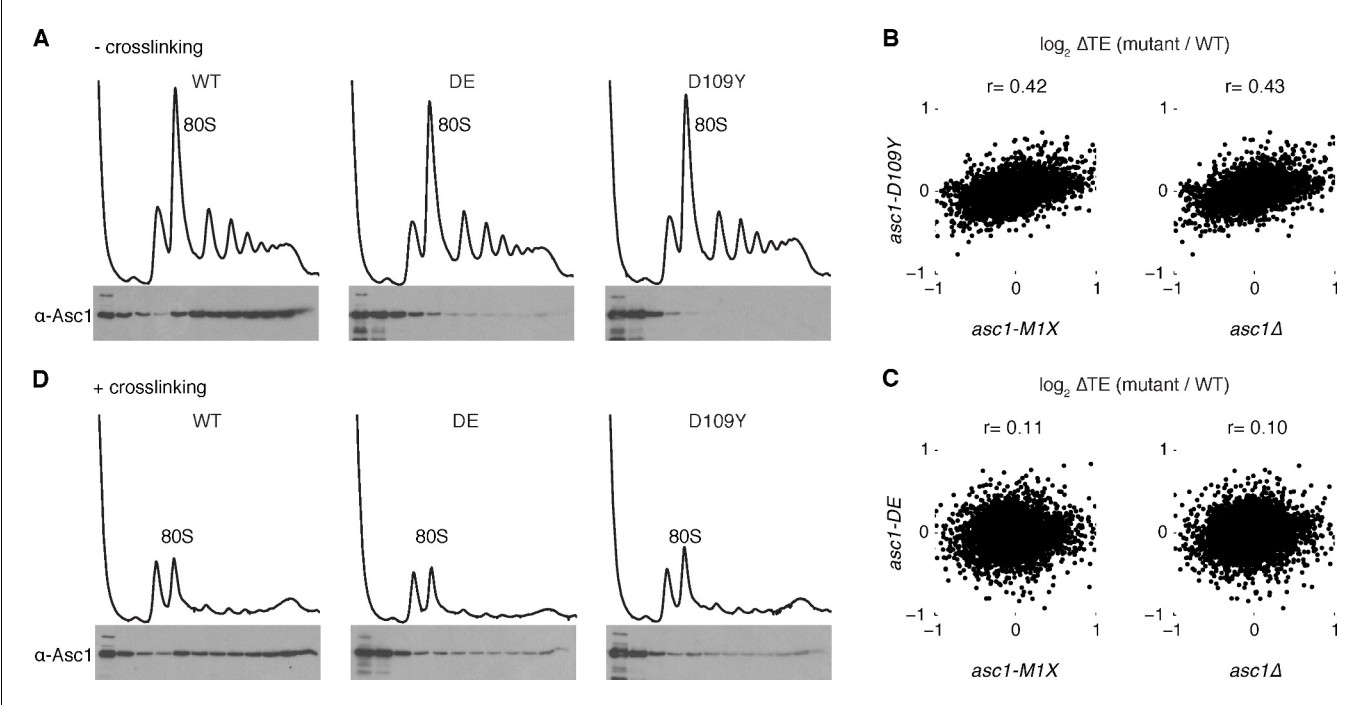

**Figure 2.** Asc1 'ribosome-binding' mutants retain ribosomal association in vivo. (**A**) Association of Asc1 mutant proteins with the ribosome assayed by Western blot of fractions isolated after velocity gradient sedimentation. (**B**, **C**) Scatterplot of TE changes between the two *ASC1* null mutants and the *asc1-D109Y* and *asc1-DE* ribosome-binding mutants. The Pearson correlation coefficients are shown. (**D**) The same as (**A**) but proteins were crosslinked with formaldehyde in vivo before sample processing.

(*Figure 1B*), and both mutations substantially decreased co-sedimentation of Asc1 with ribosomes in sucrose gradients, with D109Y having a markedly stronger effect (*Figure 2A*) that is consistent with previous reports (*Kuroha et al., 2010*).

The D109Y strong ribosome-binding mutant showed translational defects that, although correlated with those observed in the *ASC1* null mutants (r=0.43, p=10⁻²²¹ for *asc1Δ*; r=0.42, p=10⁻²⁰⁴ for *asc1-M1X*), were much smaller in magnitude (*Figure 2B*), while the DE mutant showed almost negligible effects on translation (*Figure 2C*). These findings suggest that either Asc1 primarily affects translation from a location off the ribosome, or that the ribosome-binding assay underestimates the extent of in vivo association of the D109Y and DE mutant proteins because ribosome-bound factors

**Table 1.** Properties of Asc1-sensitive mRNAs. Gene or mRNA attributes were correlated with ΔTE in the *asc1-M1X* mutant. The spearman correlation coefficients and p-values are shown.

| attribute | Spearman r (ΔTE *asc1-M1X* vs. attribute) | p-value |
|---|---|---|
| wild type protein level | 0.103 | 1.49e-2 |
| wild type translation efficiency | -0.091 | 1.55e-10 |
| tRNA adaptation index (tAI) | 0.023 | 1.17e-1 |
| 5' UTR length | -0.004 | 7.73e-1 |
| 3' UTR length | 0.079 | 9.62e-8 |
| ORF length | 0.272 | 3.05e-84 |
| 5' folding energy (MFE) | 0.030 | 3.97e-2 |
| 3' folding energy (MFE) | -0.077 | 1.83e-7 |
| poly(A) tail length | 0.029 | 7.38e-2 |

can dissociate during ultracentrifugation (*Valásek et al., 2007*). To test this second possibility, we performed formaldehyde crosslinking before ultracentrifugation. In the presence of crosslinking, we observed significant co-sedimentation of the DE and D109Y proteins with polysomes (*Figure 2D*). Crosslinking is not quantitative (*Orlando, 2000*); thus this assay underestimates the extent of ribosome binding by the mutant Asc1 proteins in vivo, which are likely much closer to wild type than previously appreciated.

An important implication of these findings is that phenotypic differences between 'ribosome-binding' alleles and *ASC1* null mutants likely reflect different degrees of perturbing ribosome function and do not constitute strong evidence for 'extra-ribosomal' activity by Asc1/RACK1. We attempted to generate stronger ribosome-binding-defective alleles by combining multiple mutations, but these proteins were expressed at very low levels potentially due to misfolding (data not shown). Given the overall correlation between *asc1-D109Y* and *ASC1* null alleles for translation changes transcriptome-wide, we infer that many of the translational changes in *asc1-M1X* and *asc1Δ* are likely to be caused by direct effects of Asc1 on ribosome function.

## Asc1 promotes translation of mRNAs with short open reading frames

To probe the mechanism by which Asc1 promotes translation of specific mRNAs, we searched for shared attributes among mRNAs with decreased TE in the *asc1-M1X* mutant. Motif analysis of 5′ UTRs revealed the presence of a U-rich sequence in mRNAs sensitive to loss of Asc1 (*Figure 3—figure supplement 1*), but not found in mRNAs resistant to loss of Asc1. However, this motif was present in only 11% of Asc1-sensitive mRNAs and so cannot be generally required for translational enhancement by Asc1. We next examined various physical properties of Asc1-sensitive mRNAs (*Table 1*). Among the tested attributes, ORF length was notably well-correlated with ΔTE in *asc1-M1X* (r=0.27, p=$10^{-84}$, *Table 1*) and ORFs <500 nts were the most strongly affected (*Figure 3A*). Short ORFs are highly translated in wild type cells (*Figure 3B* and [*Arava et al., 2003*]), an effect that has been hypothesized to reflect a higher rate of translation initiation on short mRNAs for reasons that are mechanistically mysterious (*Figure 3C* and *Arava et al., 2005*; *Shah et al., 2013*). Because short ORFs are among the most highly expressed, the loss of Asc1/RACK1 significantly alters the gene expression landscape of the cell.

We then sought to determine whether ORF length or transcript length is more predictive of translational efficiency. ORF length was slightly more predictive of wild type translation efficiency than transcript length, (*Figure 3—figure supplement 2A–E*, partial correlation r=-0.09 (p=$10^{-8}$) vs. r=0.03 (p=$10^{-1}$)). For simplicity, and because transcript boundary annotations are not available for all yeast genes, we have used the ORF length metric in subsequent analyses.

To test whether length per se, and not some other feature common to short mRNAs, is responsible for Asc1-sensitive translation, we generated two constructs with identical regulatory regions (promoter, 5′ UTR, 3′ UTR) that differed only in the length of the ORF (*Figure 3D*). These ORF length reporters contain either one or eight repeats of the I27 domain from the human cardiac protein titin, which has been used extensively in studies of protein folding because the small globular domains fold and unfold independently of each other (*Hoffmann and Dougan, 2012*). This modular architecture allows the construction of proteins of different lengths that resemble linear chains and minimizes the potential for differential protein folding or stability to impact the abundance of the reporter proteins. We performed quantitative Western blotting by fluorescent detection of a common C-terminal epitope tag in combination with qRT-PCR measurements of mRNA levels to determine the translational efficiency (protein/mRNA) of each construct (*Figure 3—figure supplement 3A*). Remarkably, the translational efficiency of the short ORF (~300 nt) was two-fold lower in the *asc1-M1X* mutant compared to the long ORF (~2400 nt) (p=0.002, *Figure 3E*). Together with the genome-wide trend, these reporter results demonstrate a role for Asc1 in the translational advantage of short mRNAs. Given that ORF length is strongly anti-correlated with translational efficiency in diverse eukaryotes (*Figure 3—figure supplement 3B–D*, data from *Guo et al., 2010*; *Stadler and Fire, 2011*), this function of Asc1/RACK1 may be conserved.

How might short ORFs be translationally privileged and sensitive to loss of Asc1? Asc1's position near the mRNA exit channel places it in close proximity to translation initiation factors that interact with the 5′ end of the mRNA during initiation, including eIF3 and eIF4G (*Kouba et al., 2012*) (*Figure 3F*, note that the structure shown is the mammalian ribosome, for which structural information regarding the orientation of eIF3 and eIF4G has been reported [*Hashem et al., 2013*;

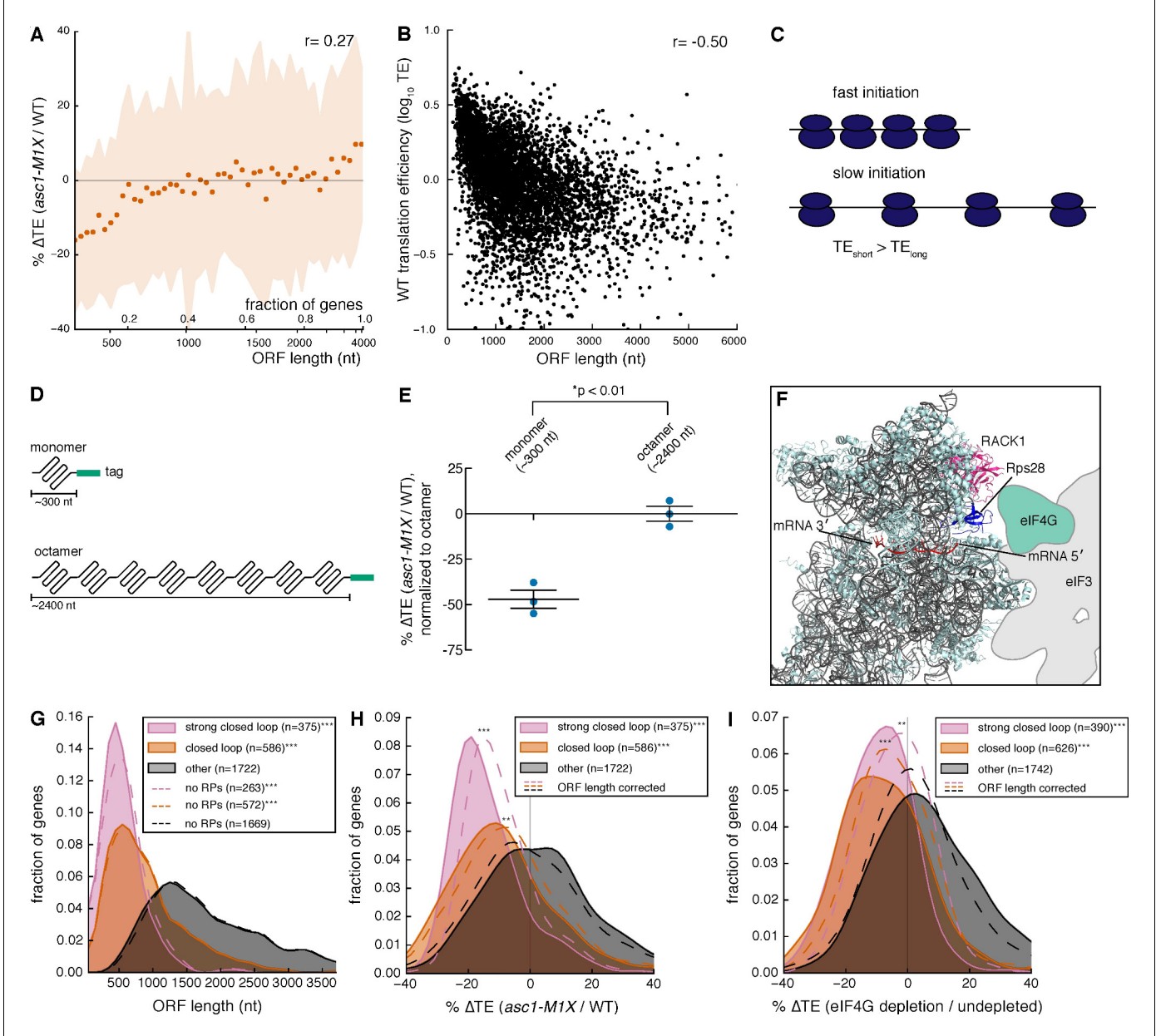

**Figure 3.** Asc1 is required for efficient translation of short ORFs that form closed loop complexes. (A) Relationship between ORF length and TE changes in *asc1-M1X*. The values shown represent the average percent change in TE for bins of 100 genes arranged by length. The ORF lengths shown correspond to the point at which the average ORF length of the bin exceeds the indicated value. Shaded areas represent +/- 1 s.d. from the average change. The *ASC1* gene is excluded from the plot. (B) Relationship between ORF length and translational efficiency in WT yeast cells (data from this study). The Spearman correlation coefficient is shown. (C) Model showing the expected effect of a higher initiation rate on short mRNAs compared to long mRNAs on translation efficiency measurements. (D) Diagram of ORF length reporter constructs. The I27 monomer was repeated to make the octamer and each ORF was fused to a C-terminal V5 epitope tag. (E) Result of ORF length reporter experiment. TE is calculated as the normalized protein (V5 tag/Pgk1) to mRNA ratio (V5 mRNA/18S) and the ΔTE (ratio between mutant and WT) is shown. Relative protein concentration was obtained from quantitative Western blotting and mRNA concentration from qRT-PCR. *p=0.002, two-tailed Student's t-test (monomer vs. octamer). Error bars are SEM from 3 biological replicates derived from independent genetic isolates of *asc1-M1X*. (F) The structure of the mammalian 48S pre-initiation complex is shown (*Lomakin and Steitz, 2013*) with the mRNA, RACK1, and Rps28, which crosslinks to the -7 and -10 positions of the mRNA relative to the AUG (*Pisarev et al., 2008*), indicated. The outline of eIF3 from *Hashem et al. (2013)* is shown. eIF4G is placed on the left arm of eIF3 based on electron microscopy data from *Siridechadilok et al. (2005)*. (G, H, I) The relationship between closed loop complex association and ORF length (p=10$^{-172}$ and 10$^{-135}$ for strong closed loop and closed loop groups vs. other mRNAs, respectively) (G), ΔTE in *asc1-M1X* (p=10$^{-71}$ and 10$^{-42}$ for strong closed loop and closed loop vs. other mRNAs, respectively) (H), and ΔTE after eIF4G depletion (p=10$^{-70}$ and 10$^{-73}$ for strong closed loop and closed loop vs. other mRNAs, respectively. Data from *Park et al., 2011*) (I). In (H) and (I), the dotted lines show the results after accounting for the relationship between ORF

*Figure 3 continued on next page*

*Figure 3 continued*

length and ΔTE using linear regression. For *asc1-M1X*, ORF length corrected p-values are $10^{-30}$ and $10^{-14}$ for strong closed loop and closed loop groups, respectively. For eIF4G depletion, ORF length corrected p-values are $10^{-17}$ and $10^{-28}$ for strong closed loop and closed loop groups, respectively. p-values are from the one-sided Mann-Whitney U test. Closed loop association groups are from *Costello et al. (2015)*. For G-I, ***p<$10^{-18}$, **p<$10^{-9}$, *p<$10^{-3}$

The following figure supplements are available for figure 3:

**Figure supplement 1.** Identification of properties of Asc1-sensitive mRNAs.

**Figure supplement 2.** Partial correlation analysis showing the relationship between wild type ORF length, transcript length, and TE.

**Figure supplement 3.** Evidence for ORF-length-dependent translational regulation.

**Figure supplement 4.** Relationship between ORF length and changes in mRNA polysome association after eIF4G depletion (data from *Park et al., 2011*).

*Lomakin and Steitz, 2013*; *Siridechadilok et al., 2005*]). eIF4G has a well-characterized role in promoting a circularized form of the mRNA in which the 5′ and 3′ regions of the mRNA are bundled together via the interaction between the eIF4G protein, associated with the mRNA 5′ cap through the eIF4F complex, and Pab1, an RNA-binding protein that binds the poly(A) tail. The mRNA in this conformation is known as the closed loop, and closed loop formation is thought to enhance translation (*Kahvejian et al., 2001*). We hypothesized that mRNAs with short ORFs might form closed loop structures more efficiently than mRNAs with longer ORFs, and that Asc1 could promote the function of the closed loop in translation.

According to this model, mRNAs with short ORFs should be more highly associated with the closed loop factors — eIF4E, eIF4G, and Pab1 — than other mRNAs. To test this prediction, we analyzed data quantifying the association of specific mRNAs with the closed loop factors and the eIF4E-binding proteins (4E-BPs) by RNA immunoprecipitation and sequencing (*Costello et al., 2015*). We grouped mRNAs into 'closed loop', 'strong closed loop', and 'other' categories based on the following enrichment profiles: 'Strong closed loop' mRNAs are enriched in immunoprecipitations of eIF4E, eIF4G, and Pab1, and de-enriched in immunoprecipitations of the 4E-BPs, which should not be associated with mRNAs in closed loops because 4E-BPs and eIF4G compete for binding to eIF4E (*Haghighat et al., 1995*). 'Closed loop' mRNAs have similar enrichment profiles to 'strong closed loop' mRNAs, but are not de-enriched for association with the 4E-BPs. Remarkably, we found that both 'closed loop' and 'strong closed loop' mRNAs were dramatically shorter than other mRNAs (median ORF lengths= 489, 774, and 1694 nt for 'strong closed loop', 'closed loop', and 'other' mRNAs, respectively). This association between ORF length and closed loop association was observed regardless of whether mRNAs encoding ribosomal proteins were included in the analysis (*Figure 3G* and *Figure 3—figure supplement 3E*). Thus, although ~30% of the 'strong closed loop' mRNAs encode ribosomal proteins, a specialized mechanism for enhancing the translation of ribosomal protein mRNAs cannot explain the ORF length bias of the 'strong closed loop' group. This discovery — that closed-loop-associated mRNAs are much shorter than other mRNAs — provides a plausible biochemical explanation for the preference for higher translation efficiency of mRNAs with short ORFs observed here and previously (*Arava et al., 2003*). Remarkably, loss of Asc1 or eIF4G depletion (data from *Park et al., 2011*) similarly decreased the translation of closed-loop-associated mRNAs (*Figure 3H and I*).

Although ORF length, closed loop enrichment, and ΔTE in *ASC1* and *eIF4G* mutants are correlated, some longer RNAs are strongly associated with the closed loop and require Asc1 for efficient translation while some short mRNAs are neither enriched with closed loop factors nor particularly dependent on Asc1 for their translation. Accounting for the global relationship between ORF length and ΔTE by linear regression showed that closed loop association has additional explanatory power for translational sensitivity to Asc1 and eIF4G: the observed reductions in translation efficiency for closed-loop-enriched mRNAs were significantly more than would be expected if ORF length alone determined their translation efficiencies (p=$10^{-14}$ and $10^{-30}$ for 'closed loop' and 'strong closed

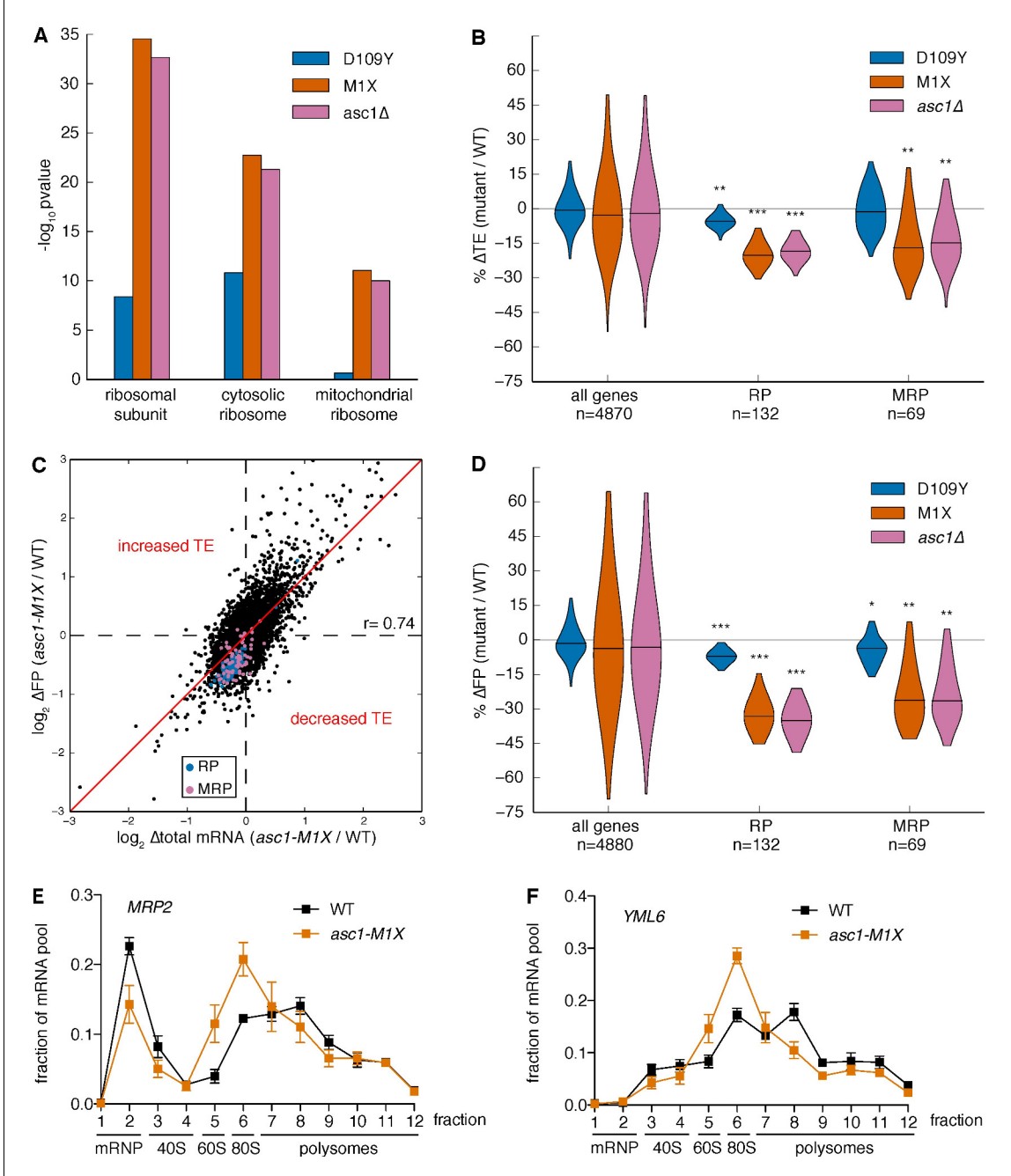

**Figure 4.** Loss of Asc1 causes decreased translational efficiency of cytoplasmic and mitochondrial ribosomal protein mRNAs. (A) GO Component category enrichments for mRNAs with decreased TE in the *ASC1* mutants. GO categories related to the top category 'ribosomal subunit' for the *asc1-M1X* mutant are displayed. (B) Violin plot showing the decreases in TE for the cytosolic ribosomal protein (RP) and mitochondrial ribosomal protein (MRP) gene sets in the *ASC1* mutants. The violin shape represents a kernel density estimation and the top and bottom of the plot extend to the most extreme data point within 1.5x of the inner quartile range. Midlines represent the medians. ***p<10$^{-18}$, **p<10$^{-9}$, *p<10$^{-3}$. (C) Scatterplot showing the decrease in both the footprint (FP) and total RNA pool for RP and MRP mRNAs. The Pearson correlation coefficient in shown. (D) As in (B), but with the change in ribosome association (FP) shown. (E, F) Polysome qRT-PCR showing decreased association of MRP genes with heavy polysomes. Values are normalized to an RNA spike-in control in each fraction and then set so that the sum of all fractions=1.

The following source data and figure supplements are available for figure 4:

**Source data 1.** GO category enrichments for mRNAs with changes in FP, total, or TE in *ASC1* mutants.

*Figure 4 continued on next page*

*Figure 4 continued*

**Figure supplement 1.** Exploring potential effects of mRNA functional categories, decay rates, and poly(A) tail length on translation efficiency measurements.

**Figure supplement 2.** Changes in mRNA levels are unlikely to explain observed translation efficiency effects in the *asc1-M1X* mutant.

loop' groups, respectively, *Figure 3H and I*). These results suggest that Asc1 is important for closed loop formation and/or stability or for closed-loop-dependent ribosome recruitment, a process that is apparently biased towards short ORFs.

What are the potential consequences of impairing translation of short mRNAs? Using gene ontology analysis, we found that transcripts annotated to the category 'ribosomal subunit' had significantly decreased TE in the *ASC1* null mutants (*asc1-M1X*, p=$10^{-35}$, *Figure 4A* and *Figure 4—source data 1*). This category is composed of short mRNAs encoding both cytoplasmic and mitochondrial ribosomal proteins (RPs, MRPs), which both displayed ~20% decreased TE in *ASC1* null mutants (*asc1-M1X*, p=$10^{-37}$ and p=$10^{-10}$; *asc1Δ*, p=$10^{-35}$ and p=$10^{-10}$, respectively, *Figure 4B*). As the median RP and MRP ORF lengths are 434 and 716 nt, respectively, their translational defects are within the range predicted by their length. Indeed, removing RP and MRP genes does not significantly alter the global relationship between ΔTE and ORF length in *asc1-M1X* (r=0.23, p=$10^{-58}$, *Figure 4—figure supplement 1A*) indicating that all classes of genes with short ORFs have decreased TE in *asc1-M1X*. Although these GO categories were the clear outliers, most GO categories with short median ORF lengths also displayed decreased TE in the *ASC1* null mutants, including several additional groups of genes whose protein products function in mitochondria (*Figure 4—figure supplement 1B*). Because short ORF length is associated with specific functional categories, loss of Asc1 — and, potentially, modulation of its activity — leads to coherent changes in gene expression.

We noted that RP and MRP mRNAs decreased in both the total RNA pool and the ribosome-protected footprint (FP) pool (*Figure 4C,D*). The additional decrease in the FP pool shows that these mRNA substrates are translationally disadvantaged in the *ASC1* mutants. In support of this interpretation, qRT-PCR analysis of polysome gradient fractions demonstrated that representative MRP mRNAs associated with fewer ribosomes in *asc1-M1X* (*Figure 4E,F*), which specifically indicates a defect in translation initiation. Because inhibiting translation initiation can induce mRNA degradation (*Coller and Parker, 2005*; *LaGrandeur and Parker, 1999*; *Schwartz and Parker, 1999*), decreased translation may account for the reduction in total mRNA levels although we cannot exclude the possibility of transcriptional effects or translation-independent effects of Asc1 on mRNA stability. We note that our translation efficiency measurements are correlated with steady-state mRNA half-life estimates using non-invasive metabolic labeling approaches (r=0.43, p=$10^{-194}$, *Figure 4—figure supplement 1C*, data from *Miller et al., 2011* and r=0.39, p=$10^{-168}$, *Figure 4—figure supplement 1D*, data from *Neymotin et al., 2014*), consistent with the hypothesis that the decay rates of mRNAs are coupled to their translational status. The same trends of decreased TE for the RP and MRP genes were observed using an rRNA-depletion strategy instead of poly(A) selection (*Figure 4—figure supplement 1E,F*), ruling out a significant effect of poly(A) tail length on our ΔTE calculations (*Subtelny et al., 2014*). Thus, Asc1 is required for efficient translation of short ORFs, which includes most ORFs encoding cytosolic and mitochondrial ribosomal proteins.

Although Asc1 has been implicated in the ribosome-dependent no-go decay pathway (*Kuroha et al., 2010*), the observed co-directional changes in mRNA abundance and translational efficiency are not consistent with widespread defects in no-go decay as a driver of changes in translation efficiency. If decreases in translation efficiency were caused by defects in no-go decay stabilizing mRNAs, thus inflating the denominator in the footprint RNA/total RNA calculation, then the levels of affected mRNAs should increase in the total RNA pool. However, the overall trend was for the levels of total mRNA for genes with decreased TE in the *asc1-M1X* mutant to go down or remain constant rather than increase (*Figure 4C*, *Figure 4—figure supplement 2A–D*).

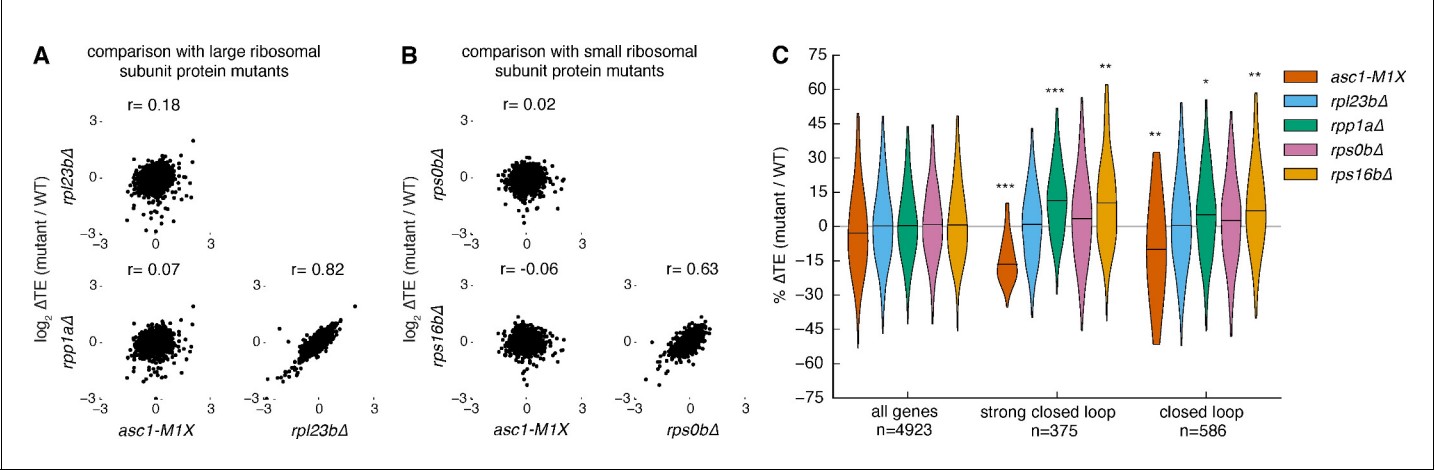

**Figure 5.** Other ribosomal protein mutants do not share translational phenotypes with the *ASC1* mutants. (A) Correlations between ΔTE among *asc1-M1X* and mutants with reduced expression of large ribosomal subunit proteins, *rpl23bΔ* and *rpp1aΔ*. The Pearson correlation coefficient is shown. (B) Correlations between ΔTE among *asc1-M1X* and mutants with reduced expression of small ribosomal proteins in the vicinity of Asc1, *rps0bΔ* and *rps16bΔ*. The Pearson correlation coefficient is shown. (C) Violin plots showing the change in TE for the 'strong closed loop' and 'closed loop' mRNAs in *asc1-M1X* and the other ribosomal protein mutants. Violin plot parameters are described in *Figure 4B*.

The following figure supplements are available for figure 5:

**Figure supplement 1.** Phenotypes of selected large ribosomal protein mutants.

**Figure supplement 2.** Ribosomal location and phenotypes of selected small ribosomal protein mutants.

## The translational defects of *ASC1* mutants are not a general consequence of perturbing the ribosome

The mRNAs that are sensitive to the loss of Asc1 are among the most efficiently translated in a cell. We therefore considered the possibility that reduced translation of these mRNAs might be a general consequence of perturbing the ribosome. To assess the specificity of the translational phenotypes of *ASC1* null mutants, we tested four additional ribosomal protein mutants, *rpl23bΔ rpp1aΔ, rps0bΔ*, and *rps16bΔ*, each of which deletes one paralog encoding a core ribosomal protein. Like *asc1-M1X*, *rpl23bΔ* and *rpp1aΔ* show reduced growth on glucose and decreased 60S subunit levels (*Figure 5—figure supplement 1A,B*). *RPS0B* and *RPS16B* encode small ribosomal subunit proteins that bind the ribosome near the mRNA exit channel in the vicinity of Asc1/RACK1 and deletion of these loci results in increased 60S subunit levels relative to 40S levels (*Figure 5—figure supplement 2A and B*). However, none of the tested ribosomal protein mutants showed notable similarity to *asc1-M1X* in their translational dysregulation genome-wide (r= -0.06 to 0.18, *Figure 5A and B*), and they did not display decreased translation efficiency of 'closed loop' mRNAs (*Figure 5C*). Because the growth and bulk translation phenotypes of these other ribosomal protein mutants are more severe than *asc1-M1X*, any shared defects in gene-specific translation should have been readily detected. Thus, decreased translation of RP genes is not a general feature of slow-growing mutants, ribosomal subunit imbalance, or perturbations in the vicinity of the mRNA exit channel near RACK1.

## Loss of Asc1 impairs mitochondrial function in yeast

To assess the physiological significance of gene-specific translation defects in *ASC1* mutants, we looked for phenotypes related to gene categories with significantly impaired translation. In particular, the requirement of Asc1 for efficient MRP translation suggested the possibility of impaired mitochondrial function in *ASC1* mutants. To assess mitochondrial health, we measured growth on the non-fermentable carbon source glycerol, which requires the activity of the mitochondrial respiratory chain to generate energy (*Dimmer et al., 2002*). When shifted to glycerol-containing media, wild type yeast resumed rapid growth after an initial adaptation phase, but the *asc1-M1X* mutant

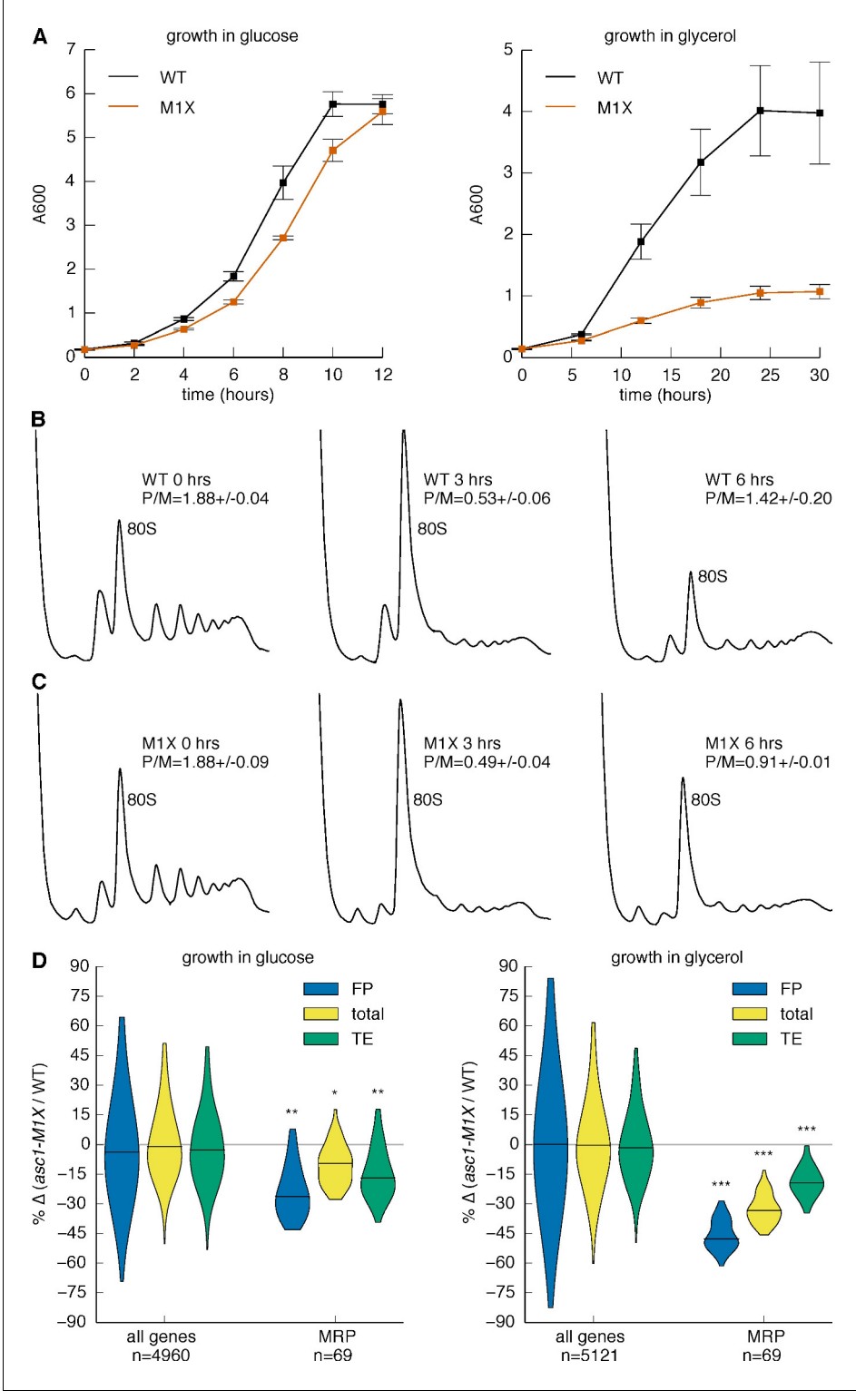

**Figure 6.** Asc1 is required for adaptation to a non-fermentable carbon source. (A) Growth curves of WT and *asc1-M1X* cells after a shift from YPAD to fresh media containing either glucose (left) or glycerol (right). Curves are averages of two biological replicates, error bars are s.d. (B,C) Polysome profiles of WT (B) and *asc1-M1X* (C) yeast after a shift from glucose- to glycerol-containing media. Yeast were shifted at OD$_{600}$=0.5. The polysome/monosome (P/M) and 60S/40S (60/40) ratios are shown with s.d. from two biological replicates. (D) Violin plot of the FP, total mRNA, and TE changes in *asc1-M1X* for MRP transcripts during growth in glucose (left, FP and TE

*Figure 6 continued*

data is also represented in *Figure 4B,D*) or after 6 hr of growth in glycerol (right). Violin plot parameters are as described for *Figure 4B*.

The following figure supplement is available for figure 6:

**Figure supplement 1.** Loss of Asc1 compromises mitochondrial function.

completed only ~3 doublings before ceasing growth (*Figure 6A*). In contrast, the *rpl23bΔ* and *rpp1aΔ* mutants grew better in glycerol than *asc1-M1X*, demonstrating the specificity of this phenotype (*Figure 6—figure supplement 1A*). Consistent with our results, a proteomic survey of *asc1Δ* cells showed a shift away from respiration and towards fermentative metabolism (*Rachfall et al., 2012*). Because mitochondrial ribosomes are required for mitochondrial biogenesis and function, it is plausible that the growth and metabolic defects of *ASC1* mutants are consequences of the translation defects observed for MRP genes.

To directly determine the impact of the MRP translation defects on mitochondrial translation activity, we performed $^{35}$S metabolic labeling assays in the *asc1-M1X* mutant in the presence of cycloheximide, which inhibits cytosolic but not mitochondrial ribosomes. Synthesis of all mitochondrially-translated proteins was reduced >two-fold in *asc1-M1X* compared to wild type (*Figure 6—figure supplement 1B and C*). Thus pervasive, moderate impairment of MRP translation is associated with substantial defects in mitochondrial protein synthesis.

Given the severe growth defects of the *asc1-M1X* mutant in glycerol, we wondered whether the moderate impairment of MRP translation observed in glucose would worsen under conditions of increased MRP expression. Adaptation to growth in non-fermentable carbon sources or low glucose is accompanied by a rewiring of the transcriptional network in yeast (*Galdieri et al., 2010*) and widespread reprogramming of translation (*Vaidyanathan et al., 2014*). To investigate whether the glycerol growth defect of *asc1-M1X* is linked to inadequate translational adaptation, we examined translation genome-wide 6 hr after transfer from glucose to glycerol, a point just before the resumption of rapid growth in wild type cells (*Figure 6A*) and coincident with the recovery of polysomes after the initial collapse upon glucose withdrawal (*Figure 6B*). In the *asc1-M1X* mutant, polysomes recovered only partially (*Figure 6C*). Moreover, the ribosome-associated pool was strongly depleted of MRP mRNAs compared to wild type (*Figure 6D*). However, the magnitude of translational defect (ΔTE) for this class of mRNAs was similar in both glucose and glycerol, supporting a constitutive rather than regulatory role for Asc1 in translation of MRP mRNAs (*Figure 6D*). The fact that *asc1-M1X* shows a bulk translation defect in glycerol but not in glucose may reflect the fact that MRP mRNAs make up a larger portion of the translatome in glycerol. Thus, the cellular context is an important factor in determining the phenotypic consequences of translational perturbations in *asc1-M1X* and likely in other ribosomal protein mutants as well. Taken together, our results suggest an important role for Asc1 in supporting cellular respiration by promoting synthesis of mitochondrial ribosomal proteins.

## Discussion

Here we have demonstrated that the eukaryote-specific ribosomal protein Asc1/RACK1 is required for efficient translation of short mRNAs, a category that includes functionally related groups of genes required for vital cellular processes (e.g. cytoplasmic and mitochondrial ribosomal proteins). A correlation between ORF length and translation efficiency or ribosome density has been observed since the advent of genome-wide translation profiling (*Arava et al., 2003*; *Ingolia et al., 2009*) and we observed this relationship in data collected from diverse eukaryotes including yeast, nematodes, mice, and humans (*Guo et al., 2010*; *Stadler and Fire, 2011*). To account for this trend, it was proposed that the rate of translation initiation is higher for short ORFs (*Arava et al., 2005*; *Shah et al., 2013*), but the mechanism(s) underlying length-dependent initiation rate differences were unknown.

It has been suggested that the increased probability of mRNA circularization by diffusion could make initiation more efficient on short mRNAs (*Chou, 2003*; *Guo et al., 2015*). Our results add an additional nuance to these physical models — the presence of a ribosome-dependent regulatory

mechanism that specifically enhances the translation of short mRNAs by promoting the formation and/or function of the closed loop. Our analyses reveal a clear trend that short mRNAs preferentially associate with closed loop factors in vivo, and consistent with these observations, short mRNAs form more stable closed loop complexes than longer mRNAs in vitro (*Amrani et al., 2008*). A challenge for the future will be to determine how the mRNA, the closed loop factors, and the ribosome cooperate to privilege the translation of short mRNAs.

How might Asc1/RACK1 promote closed loop formation? RACK1's placement on the solvent exposed side of the head of the small subunit puts it in close proximity to the mRNA exit channel, in a position with the potential to interact with the mRNA-bound closed loop factors during initiation. Intriguingly, eIF4G co-purifies with Asc1 from yeast lysates under stringent conditions in which most other initiation factors do not (*Gavin et al., 2002*; *2006*), suggesting that Asc1 may interact directly with the closed loop via eIF4G. Our results also raise the possibility that translation of many mRNAs could be co-regulated by mechanism(s) that target Asc1/RACK1's function in closed-loop-dependent initiation. Moving forward, it will be important to determine how the many signaling pathways that have been linked to Asc1/RACK1 impact the translation of closed-loop-dependent mRNAs.

More generally, our study highlights the fact that individual ribosomal proteins can contribute to efficient translation of subsets of mRNAs with important consequences for cellular physiology. In particular we show that loss of the non-essential ribosomal protein Asc1/RACK1 causes a concerted decrease in MRP expression that leads to mitochondrial insufficiency. Given the central role of mitochondria in energy and metabolite production in eukaryotic cells, it is not surprising that mitochondrial defects elicit pleiotropic consequences (*Calvo and Mootha, 2010*; *Fleming et al., 2001*; *Kushnir et al., 2001*; *Shoffner et al., 1990*; *Vafai and Mootha, 2012*). In light of our findings, many of Asc1/RACK1's ascribed cellular functions should be re-evaluated for potential connections to mitochondrial dysfunction. Finally, it is intriguing that several distinct mutations in human ribosomal proteins and ribosome biogenesis factors result in anemia, the cause of which is currently the source of much debate (*Freed et al., 2010*; *Narla et al., 2011*). Given that many forms of heritable anemia have been traced to defects in mitochondrial iron metabolism (*Dailey and Meissner, 2013*; *Huang et al., 2011*), it will be interesting to see whether translation of nuclear-encoded mitochondrial proteins is affected in these diseases and whether these defects contribute to pathogenesis.

## Materials and methods

### Plasmid construction

The cDNA encoding the I27 domain monomer from human cardiac titin was a generous gift from Julio Fernandez. The I27 monomer was fused to a serine-glycine linker (SGGGGG) followed by the V5 epitope tag. The I27 octamer was made using the iterative subcloning method that relies upon the compatible cohesive ends of BamHI and BglII and results in an arginine-serine linker added between individual domains (*Hoffmann and Dougan, 2012*). I27 proteins were expressed under the *GAL1* promoter and followed by the *CYC1* terminator in the pRS415 low-copy yeast vector.

### Yeast strain construction

Deletion strains of the *ASC1, RPL23B*, and *RPP1A* loci were obtained by homologous recombination using the pFA6a-kanMX6 plasmid as a template and PCR product adding 40 nt of homology to each side of the kanMX6 cassette (*Longtine et al., 1998*). Isolates were confirmed by PCR. Deletion strains of *RPS0B, RPS16B*, and their isogenic wild type were obtained from the Sigma1278b deletion collection (*Dowell et al., 2010*). To make the *ASC1* protein null alleles, a codon early in the *ASC1* open reading frame was mutated to a stop codon, denoted as X (i.e. M1X, E5X). Integration of mutant *ASC1* alleles was performed using the two-step gene replacement strategy. First, the *URA3* marker was integrated at the *ASC1* locus. Subsequently, *ASC1* mutant alleles were amplified by PCR from plasmid templates and integrated into the asc1::URA3 strain at the *ASC1* locus. Isolates were identified by 5-FOA resistance and correct integration was confirmed by sequencing. All strains were constructed in the Sigma1278b strain background.

## Yeast growth

Yeast were cultivated in liquid or on solid (2% agar) YPAD media (yeast extract, peptone, dextrose (2% w/v) supplemented with adenine hemisulfate). Liquid cultures were grown with rapid agitation at 30°C, unless otherwise noted, and harvested at OD 0.6–0.9 (0.6-0.7 for ribosome footprint profiling experiments in YPAD). For glycerol shift polysome experiments, yeast were grown to mid log phase (OD 0.5–0.6) in YPAD and then media was removed by brief centrifugation and replaced with YPAG (YPA + 3% (w/v) glycerol). For the yeast growth curves, yeast were diluted from saturated cultures into fresh media and allowed to double 1–2 times before rediluting to an OD of 0.1 in glucose- or glycerol-containing media.

## Polysome analysis

Cycloheximide (CHX, Sigma-Aldrich, St. Louis, Missouri) was added to a final concentration of 0.1 mg/ml to cells and incubated an additional 2 min at the growth temperature with shaking. Cells were rapidly cooled and washed with polysome lysis buffer (PLB: 20 mM Hepes-KOH, pH 7.4, 2 mM Mg acetate, 100 mM K acetate, 3 mM DTT, 0.1 mg/ml CHX + 1% Triton X-100). Formaldehyde crosslinking experiments were performed as described (Valásek et al., 2007). 10–15 OD260 units were loaded on 10–50% sucrose gradients in polysome gradient buffer (PGB: PLB –Triton) and centrifuged in an SW 41 rotor (Beckman Coulter, Brea, California) at 35,000 rpm for 3 hr. Fractions were collected from the top using a BioComp Gradient Station (Biocomp Instruments, Canada). To calculate the ratio of free 60S/40S subunits, A254 traces of the native polysome profiles (without dissociation into free subunits) were quantified with a custom script, available on github: https://github.com/marykthompson/Thompson_eLife_2016/. Minima were identified and used as boundaries for each peak. Values are the integral under the curve to the baseline, which was set as a line connecting the lowest minimum in the first half of the trace with the lowest minimum in the second half of the trace.

## Ribosome footprint profiling

Ribosome footprint profiling was performed essentially as described (Ingolia et al., 2009) with the following modifications: monosomes were isolated manually from 10–50% sucrose gradients. 50 A260 units were digested with 750 U of RNAse I (Ambion, Waltham, Massachusetts). Selective precipitation was used to enrich for small RNA fragments prior to size selection of 28mers on denaturing gels. In brief, RNA samples were resuspended in GuHCl buffer (8 M guanidine HCl, 20 mM MES hydrate, 20 mM EDTA) and brought to 33% ethanol before binding to a silica-based column (Zymoprep-V, Zymoresearch, Irvine, California) to precipitate and remove large RNAs. The eluate was brought to 70% ethanol to precipitate small RNAs. Total RNA for accompanying RNA-seq samples was isolated from the same cell extracts used for footprint library generation using the hot acid phenol method. Poly(A) selection was performed using oligo-dT cellulose (Sigma-Aldrich or NEB, Ipswich, Massachusetts) as previously described (Sambrook et al., 2001). For experiments using rRNA-depletion to enrich for coding transcripts, the Ribo-Zero kit (Epicentre, Madison, Wisconsin) was used. The asc1-DE and matched WT libraries were constructed using an earlier version of the protocol that used Microcon YM-100 (EMD Millipore, Billerica, Massachusetts) filters to enrich for small RNA fragments, poly(A) tailing to capture the small RNA fragments, and downstream library construction steps as previously described (Ingolia et al., 2009). For all other libraries, we ligated a preadenylated 3' adaptor (5Phos/TGGAATTCTCGGGTGCCAAGG/3ddC/) to the fragments using T4 RNA Ligase 1 (NEB). First strand synthesis was performed with Superscript III (Life Technologies, Carlsbad, California) or AMV (Promega, Madison, Wisconsin) using primer OJA225 (/5Phos/GATCGTCGGACTGTAGAACTCTGAACCTGTCGGTGGTCGCCGTATCATT/iSp18/CACTCA/iSp18/GCCTTGGCACCCGAGAATTCCA). cDNA was amplified using primer oNTI230 (AATGATACGGCGACCACCGA) and (CAAGCAGAAGACGGCATACGAGATXXXXXXGTGACTGGAGTTCCTTGGCACCCGAGAATTCCA), where XXXXXX denotes a six nucleotide barcode used to distinguish samples run in the same lane. Samples were run on an Illumina HiSeq 2000 instrument or an Illumnina GAIIx.

## qRT-PCR

RNA was extracted using the hot acid phenol method. RNA was treated with TURBO DNase (Life Technologies). First strand synthesis was performed with AMV Reverse Transcriptase (Promega) using an anchored oligo-dT primer (for coding transcripts) or a random hexamer primer (for SNR24).

Quantitative PCR was performed with SYBR Fast reagents (Kapa Biosystems, Wilmington, Massachusetts) using a Lightcycler 480 (Roche, Switzerland). Gene-specific primer sequences are: ACT1: (TTC TGAGGTTGCTGCTTTGG, CTTGGTGTCTTGGTCTACCG), ASC1: (ATGTTTGGCCACTTTGTTGG, GTTACCGGCAGAAATGATGG), MRP2: (AATAGGTGCGTGGACTCTGG, CTGGCAAATTACCCTTCAGAGC), SNR24: (TTGCTACTTCAGATGGAACTTTG, TCAGAGATCTTGGTGATAATTGG), V5: (AGATCTTCCGGAGGCGGG, GGATCTATTACGTAGAATCGAGACC), YML6: (AGAGTAGGCGCCTCAAATCC, TTGGAGAGTTAGCATCCCCG), 18S: (TGGCGAACCAGGACTTTTAC, CCGACCGTCCCTATTAATCAT), FLUC: ( GTACCAGAGTCCTTTGATCGTGA, ACCCAGTAGATCCAGAGGAATTC).

## Western blotting

Total protein levels were determined using the BCA assay (Thermo Scientific, Waltham, Massachusetts). For total Asc1 level quantification, 1 μg of total protein obtained by TCA precipitation followed by cell lysis was loaded onto 12% SDS-PAGE gels. For polysome Westerns, the same volume of each fraction was loaded per well. Blots were overexposed to show the remaining ribosome-associated protein for the ribosome-binding mutants. Membranes were blotted with α-Asc1 (*Coyle et al., 2009*) and α-Pgk1 (Life Technologies 22C5D8). After secondary antibody incubation, blots were incubated with ECL (GE Healthcare Life Sciences, United Kingdom) and exposed to X-ray film.

## ORF length reporter assays

Yeast grown overnight in SC-Leu (synthetic complete media lacking leucine) were diluted to OD 0.2 in YPA + 2% galactose and grown for 8 hr before harvest. Cells were lysed in PBS pH 7.4 supplemented with protease inhibitors (1X Roche complete mini EDTA-free, 1 mM PMSF) with glass beads. Total protein was quantified by the BCA assay (Thermo Scientific) and 1 ug (octamer) or 2 ug (monomer) total protein was loaded per lane with each sample loaded in 4 different lanes as technical replicates for each of three biological replicates. A standard curve encompassing 2X, 1X, 0.5X and 0.25X of the WT extract concentration was loaded on each gel. Western blotting was performed using the ECL Plex kit (GE) according to the manufacturer's instructions and blots were scanned with a Typhoon imager (FLA 9500, GE). Primary antibodies were α-Pgk1 (Life Technologies 22C5D8) and α-V5 (Sigma-Aldrich V8137). Images were quantified with ImageStudio (LI-COR Biosciences, Lincoln, Nebraska) and the values of each sample were calculated relative to the standard curve. Although all standards were in linear range (linear fit of signal vs. concentration $r^2 \geq 0.95$ for all blots), we used a quadratic fit as it fit the standards slightly better. RNA was extracted from the extracts in parallel, and the mRNA levels of each reporter were quantified by qRT-PCR using primers recognizing the region encoding the V5 tag and normalized to 18S levels also determined by qRT-PCR. For each sample, a translation efficiency was calculated from the ratio of the normalized protein levels of the reporter (V5 protein/Pgk1 protein) to the normalized mRNA levels of the reporter (V5 mRNA/18S rRNA).

## Mitochondrial translation

Mitochondrial translation products were labeled with $^{35}$S-methionine as previously described (*Funes and Herrmann, 2007*). In brief, cells were grown overnight in SC-Met (with glucose) to OD 0.4 then transferred to SC-Met with glycerol for 3 hr. Equal OD units of cells were then incubated with $^{35}$S-methionine (EasyTag L-[35S]-Methionine, PerkinElmer, Waltham, Massachusetts) and cycloheximide to inhibit cytoplasmic protein synthesis. After 30 min, total protein synthesis was halted by the addition of puromycin. TCA-precipitated protein was visualized by Coomassie staining (total protein normalization) and autoradiography on a Typhoon imager (mitochondrial proteins). Total protein in each sample was quantified with ImageJ using Coomassie signal across the whole lane. Six bands corresponding to mitochondrial translation products were quantified with ImageQuant (GE).

## Read mapping and positional alignment

Yeast reads were aligned to the Sigma1278b (*Dowell et al., 2010*) genome downloaded from the Saccharomyces Genome Database on June 29, 2014. We used Tophat to map first to annotated splice junctions and then to the genome. We used only uniquely-mapping reads for all downstream analyses. Because ribosome footprint reads generally start 12 nt upstream of start codons and end

18 nt upstream of stop codons (*Ingolia et al., 2009*), we used only reads mapping within these boundaries. Additionally, to avoid potential variability that can be present at the 5' end of mRNAs, we excluded the first 30 codons from counting for quantification of gene expression.

## Gene expression analysis

For comparisons between libraries, we used normalized values obtained from running count data through the DE-Seq package (*Anders and Huber, 2010*) because RPKM values are strongly biased by the transcript lengths of the RNA pool (*Wagner et al., 2012*). For gene expression measurements, we only included genes for which there were at least 128 mapping reads total among the libraries used for the analysis (*Ingolia et al., 2009*). All analyses were performed with custom Bash and Python scripts written in-house, available on github: https://github.com/marykthompson/Thompson_eLife_2016/. Data in figures represent the average of two biological replicates. Figures were constructed using Matplotlib (*Hunter, 2007*).

## ORF length correction of ΔTE values for closed loop groups

To determine whether the decrease in translation efficiency of the 'closed loop' mRNAs in *asc1-M1X* could be accounted for completely by the relationship between ΔTE and ORF length, we first regressed ΔTE against ORF length. We then took the residuals from this correlation (i.e. the part of ΔTE that cannot be accounted for by the global correlation between ΔTE and ORF length) and plotted these values among the 'strong closed loop', 'closed loop' and 'other' mRNAs, as shown by the dashed lines in *Figure 3H and I*. Note that this analysis assumes linear relationships between ΔTE and ORF length. These results demonstrate that the decrease in translation efficiency of the 'closed loop' mRNAs in *asc1-M1X* is more than would be expected if ΔTE was determined entirely by ORF length, thus suggesting that 'closed loop' enrichment may be more important. However, as with all correlative analyses, the results cannot assign causality.

## Analysis of TE in other organisms

For correlations of TE with ORF length shown in *Figure 3—figure supplement 3*, processed data files were downloaded from NCBI GEO and used to calculate TE. Only genes in which the pooled the reads or scaled reads (for *Stadler and Fire, 2013*) from footprint and total RNA libraries reached 128 reads were included.

## Pathway analysis

To identify groups of genes with significantly altered TE in yeast mutants, we used the Mann Whitney U test and report one-sided p-values for groups of genes with significantly altered TE each condition. For this analysis, we included all genes without filtering for read cutoff and added a pseudocount of one read in cases where >1 read was detected in some but not all libraries.

## Motif finding

We used MEME (*Bailey and Elkan, 1994*) to identify motifs present in 5' UTRs of the selected groups of mRNAs. 5' UTR boundaries were taken from the median UTR length reported in *Pelechano et al. (2014)*. UTRs <8 nt were excluded from the motif analysis. WebLogo (*Crooks et al., 2004*) was used to generate sequence logos.

## Data sources for mRNA attributes

5' and 3' UTR lengths were taken as the median length from *Pelechano et al. (2014)*. MFEs were calculated by running these sequences through RNAfold (*Gruber et al., 2008*) with temperature set to 30°C and otherwise default parameters. Translation adaptation index values per gene were calculated by Eckhard Jankowsky and colleagues using values from *Tuller et al. (2010)*. Poly(A) tail length was taken from *Subtelny et al. (2014)*. Wild type protein levels were taken from *de Godoy et al., 2008*.

## Acknowledgements

We thank N Ingolia, J Weissman, D Bartel, and members of the Gilbert lab for discussion; and B Zinshteyn and P Vaidyanathan for discussion and scripts. The sequencing was performed at the Bio-Micro Center under the direction of S Levine. This work was supported by the National Institutes of Health (GM094303) to WVG. This work was supported in part by the NIH Pre-Doctoral Training Grant T32GM007287.

## Additional information

### Funding

| Funder | Grant reference number | Author |
| --- | --- | --- |
| National Institutes of Health | GM094303 | Mary Katherine Thompson<br>Maria Fernanda Rojas-Duran<br>Paritosh Gangaramani<br>Wendy V Gilbert |
| National Institutes of Health | T32GM007287 | Mary Katherine Thompson<br>Paritosh Gangaramani |

The funders had no role in study design, data collection and interpretation, or the decision to submit the work for publication.

### Author contributions

MKT, Performed ribosome profiling, Performed most experiments, Conception and design, Acquisition of data, Analysis and interpretation of data, Drafting or revising the article; MFR-D, Performed ribosome profiling, Acquisition of data; PG, Characterized translation and growth of rps mutants, Acquisition of data; WVG, Conception and design, Analysis and interpretation of data, Drafting or revising the article

### Author ORCIDs

Mary K Thompson, http://orcid.org/0000-0002-4947-6048
Paritosh Gangaramani, http://orcid.org/0000-0002-4893-8167
Wendy V Gilbert, http://orcid.org/0000-0003-2807-9657

## Additional files

### Major datasets

The following dataset was generated:

| Author(s) | Year | Dataset title | Dataset URL | Database, license, and accessibility information |
| --- | --- | --- | --- | --- |
| Mary K. Thompson, Maria F. Rojas-Duran, Paritosh Gangaramani, Wendy V Gilbert | 2016 | The ribosomal protein Asc1/RACK1 is required for efficient translation of short mRNAs | http://www.ncbi.nlm.nih.gov/geo/query/acc.cgi?acc=GSE61753 | Publicly available at NCBI Gene Expression Omnibus (accession no. GSE61753) |

The following previously published datasets were used:

| Author(s) | Year | Dataset title | Dataset URL | Database, license, and accessibility information |
| --- | --- | --- | --- | --- |
| Guo H, Ingolia NT, Weissman JS, Bartel DP | 2010 | Mammalian microRNAs predominantly act to decrease target mRNA levels | http://www.ncbi.nlm.nih.gov/geo/query/acc.cgi?acc=GSE22004 | Publicly available at NCBI Gene Expression Omnibus (accession no. GSE22004) |

| | | | | |
|---|---|---|---|---|
| Stadler M, Fire A | 2013 | mRNA and Ribosome Profiling in Four Nematode Species Traversing a Shared Developmental Transition | http://www.ncbi.nlm.nih.gov/geo/query/acc.cgi?acc=GSE48140 | Publicly available at NCBI Gene Expression Omnibus (accession no. GSE48140) |
| Park E, Zhang F, Warringer J, Sunnerhagen P, Hinnebusch AG | 2010 | Depletion of eIF4G from yeast cells narrows the range of translational efficiencies genome-wide | http://www.ncbi.nlm.nih.gov/geo/query/acc.cgi?acc=GSE25721 | Publicly available at NCBI Gene Expression Omnibus (accession no. GSE25721) |
| Costello J, Castelli L, Rowe W, Kershaw CJ, Sims P, Grant CG | 2014 | Global assessment of the closed loop components (eIF4E, eIF4G and PABP) and the translational repressors (4E-BPs) in mRNA recognition for translation initiation. | https://www.ebi.ac.uk/arrayexpress/experiments/E-MTAB-2464/ | Publicly available at ArrayExpress (accession no. E-MTAB-2464) |

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
