## [Decision Letter]

Thank you for submitting your work entitled "Asc1/RACK1 is required for efficient translation of short mRNAs" for peer review at *eLife*. Your article has been favorably evaluated by James Manley (Senior editor) and three reviewers, one of whom, Allan Hinnebusch, is a member of our Board of Reviewing Editors, and another is Leos Valasek.

The reviewers have discussed the reviews with one another and the Reviewing editor has drafted this decision to help you prepare a revised submission.

Your paper has been reviewed by three experts in the field and the general consensus is that it is potentially suitable for publication pending the outcome of additional analyses and revisions.

One key issue concerns the view that there is inadequate analysis of protein synthesis from native genes or reporters to back up the claim that translation of short mRNAs with a heightened propensity to form the closed-loop is particularly sensitive to elimination of Asc1. The one reporter you analyzed where the coding sequence length was varied systematically was considered a creative approach, but the results obtained were not very compelling. Suggestions were offered for a different kind of reporter analysis, where a native long mRNA would be systematically shortened, and to examine protein synthesis of a panel of native mRNAs representing Asc1-dependent versus independent genes to confirm the interpretations of the ribosome-profiling data for authentic mRNAs. You could extend the analysis in Figure 4 to include additional Asc1-dependent mRNAs, and also include Asc1 independent mRNAs, but it might be difficult to detect changes in polysome distribution for short mRNAs, in which case measurements of protein and mRNA synthesis or reporter expression would be required. I recall that in our paper on eIF4G depletion (Park et al), which you used creatively in your analysis, we validated a number of mRNAs as being eIF4G-dependent or -independent, and perhaps some of these mRNAs that are short and enriched for both eIF4G and PABP occupancy could be analyzed in your strains to achieve stronger validation of your ribosome profiling data.

Another important issue was whether you have effectively untangled the contributions of mRNA length and propensity to form the closed loop as determinants of Asc1-dependence using bioinformatics and statistical analysis. There was general agreement among the reviewers during the consultation session that you should attempt to bolster your model by showing that an existing eIF4G mutation known to disrupt the closed loop mRNP would dampen the effects of eliminating Asc1 on translational efficiency of Asc1-dependent mRNAs or reporters. As these eIF4G mutations were made in the S288C background, such experiments would also address a concern that your study employed the Sigma background, which is rarely employed in studies on gene expression. A related point is that, without demonstrating that elimination of at least one other 40S protein in the vicinity of Asc1 on the solvent-exposed surface of the 40S subunit does not have the same effects on TE you observed on elimination of Asc1, it is dangerous to conclude that Asc1 is specifically dedicated to this regulatory function.

There was also agreement that it is necessary to measure mitochondrial protein synthesis and show that it is reduced in cells lacking Asc1 rather than relying on growth assays in nonfermentable carbon sources. And one of the reviewers is concerned that your conclusion that Asc1 is required for normal 60S biogenesis is at odds with other careful work done previously that ruled out this possibility, and feels that your measurements of 40S:60S subunit ratios from the polysome profiles in Figure 1 do not represent the most rigorous approach to establishing this point. I think it is important to carefully consider these criticisms to avoid "muddying the waters" concerning the involvement of Asc1 in ribosome biogenesis.

Another of the reviewers was justifiably concerned that you may have overlooked the effects of deleting Asc1 on "no-go" decay and that defects in this pathway and attendant changes in progression of stalled ribosomes on no-go substrates, or changes in mRNA levels, could be influencing conclusions about alterations of translational efficiency. It is important to consider this possibility and more fully document the effects of eliminating Asc1 on mRNA levels and the extent to which changes in TE might be driven by defects in no-go decay.

Finally, there were some issues with a lack of adequate descriptions of procedures and bioinformatics analysis.

*Reviewer #1:*

This paper shows convincingly that elimination of Asc1 alters the translational efficiency of a substantial fraction of the yeast translatome with a heightened effect on short mRNAs. As short mRNAs tend to be the most efficiently translated, Asc1 is thus particularly important for the group of most highly translated mRNAs in the cell. Analysis of synthetic reporter constructs in which mRNA length was progressively increased suggests that short length per se confers an increased dependence on Asc1; however, unfortunately the data in Figure 3 have a lot of variance and the results are significantly different only for the longest and shortest mRNAs in the series of constructs. As it was shown previously by Jacobson that short mRNAs can more readily form the closed-loop intermediate in vitro, they attempt to explain the Asc1-dependence of short mRNAs by analyzing a published data set on eIF4G and PABP binding to yeast mRNAs and find that (in WT cells) short mRNAs are indeed enriched in the mRNA classes with high occupancies of both proteins and thus likely to form the closed-loop mRNP intermediate. These mRNAs are also particularly dependent on Asc1, but I think that this would be expected from the finding that short mRNAs are Asc1-dependent. They claim to have shown that Asc1-dependence of the "strong" closed-loop category can be functionally separated from the Asc1 dependence of short mRNAs, but it's not clearly explained how they accomplished this, nor whether it can be done convincingly with only bioinformatics. They conclude that Asc1 is either important for closed-loop formation, or for the enhanced translation (ribosome recruitment?) of mRNAs in the closed-loop configuration. They finish by showing that the reduced translation of short mRNAs encoding mitochondrial ribosomal proteins is associated with a growth defect of *asc1* cells in non-fermentable carbon sources where mitochondrial function is important. However, they do not show directly that mitochondrial protein synthesis is impaired in *asc1* cells and it seems that they cannot rule out effects on expression of other genes required for growth in nonfermentable carbon sources.

General critique:

It is quite interesting that Asc1 is found to be particularly important for efficient translation of short mRNAs and also mRNAs with strong potential for closed-loop formation, and it is noteworthy that they have discovered in previously published datasets unsuspected relationships between short mRNA length and both high closed-loop forming potential and heightened dependence on eIF4G. However, since short mRNAs are enriched for both closed-loop formation potential and for dependence on eIF4G, it isn't entirely clear whether it is length per se, ability to form the closed loop, or dependence on eIF4G that confers the greater than average requirement for Asc1 for efficient translation.

Specific comments:

The key experiment in Figure 3 should be bolstered with additional replicates in an effort to increase the statistical significance of differences between the monomer vs dimer or trimer.

If that can't be accomplished, then an additional, complementary experiment could be conducted where they start with a native, long mRNA that is Asc1-independent and show that Asc1-dependence is conferred by progressive removal of coding sequences. Perhaps this could be done for an mRNA whose protein product is a scaffold comprised of multiple modular domains.

An important experiment that would test their conclusion that Asc1 is important for enhanced translation of mRNAs in the closed-loop configuration would be to examine the effect of deleting Asc1 in a previously published strain in which the PABP binding site in eIF4G1 is deleted, eliminating its ability to form the closed -loop intermediate, and eIF4G2 is deleted. (Tarun and Sachs showed in 1997 that the entire N-terminus of eIF4G1 can be deleted in a strain lacking eIF4G2 with only moderate effects on cell growth.) Their model strongly predicts that the effect of deleting Asc1 on translation of short, "strong closed-loop mRNAs" will be substantially dampened in this eIF4G1 mutant.

They do not show directly that mitochondrial protein synthesis is impaired in *asc1* cells and cannot rule out effects on expression of other genes required for growth in nonfermentable carbon sources. It seems necessary to measure mitochondrial protein synthesis directly in *asc1* cells, which is a feasible experiment, to justify their claims.

*Reviewer #2:*

In this manuscript the authors assess the role of the ASC1/RACK protein in protein synthesis. They use the ribosome footprinting technology to address the impact of different mutants in *ASC1* and conclude that these mutations preferentially alter the translation of short mRNAs. I found the data interesting but oversold in places. The paper was also quite difficult to read due to a lack of technical detail in many of the methods descriptions and legends. Finally, I was not convinced that the authors had effectively shown that Asc1 affected translation directly.

The authors desperately need to provide some direct measure of protein synthesis in this paper. The ribosome footprinting pattern across mRNAs provides an approximation of translation and how this changes, but there are other reasons that ribosomes might accumulate on mRNAs (see point about no-go mRNA decay below). So claims are made about the rates of initiation and the role of Asc1 in protein synthesis without actually ever measuring the synthesis rate of any endogenous proteins directly. In Figure 3, the authors do measure the translation efficiency of a series of artificial engineered proteins to study the correlation between ORF length and the impact of *ASC1* mutation. However, they need to show the primary data here so that the impact of the mutation on protein synthesis can be directly assessed by the reader. Also in my view validation that Asc1 specifically affects the synthesis of protein from some short mRNAs relative to some longer ones is essential.

The choice of strain for this study is very strange. While the Σ1278b strain is widely used for studies into filamentous growth in *S. cerevisiae* it is quite rarely used outside this, and it has some odd characteristics that raise concerns over the generality of the results shown. As has been shown previously on a number of occasions and is now shown in the polysomes presented here, Σ1278b is very unusual in that it has abnormally high levels of free 60S subunits. The *asc1* null has previously been shown to decrease 60S subunit levels and again the authors reconfirm this observation. The authors state that the *ASC1* mutants have relatively subtle effects on protein synthesis and make statements about how subtle effects on translation as judged by polysome profiling can conceal a wider perturbation of cellular translation. It is possible that the abnormal ratio of free 40S/60S subunits in the Σ strain obviates potentially more dramatic effects of the *ASC1* mutation on translation. In my view the authors should really test key observations from their study in a laboratory strain of yeast that has a more standard profile of free 40S and 60S subunits. Obviously this may be an unrealistic expectation as ribosome footprinting is neither cheap nor trivial- but some attempt to generalize the data could be made and the non-standard nature of the strain and the ratio of ribosomal subunits should be commented upon.

For the polysomes presented in Figure 1 the polysomes profiles have very high 80S peaks relative to the polysome peaks for unstressed cells, this suggests that some stress has occurred during the polysome preparation? Could this be masking the impact of the *asc1* mutations on polysome association?

Figure 2 – I find this formaldehyde polysome analysis particularly unconvincing. The polysomes are barely visible and the pattern of the other peaks looks substantially altered. The authors claim that Asc1 is maintained across the gradient in the mutants seems inappropriate based on the data presented and based on what is in the literature.

As the authors state, Asc1 has also been shown by the Inada group to play a role in the no-go mRNA decay process. It promotes translation arrest to facilitate mRNA decay by this pathway. It is therefore possible that the alterations in ribosome density on mRNAs for *ASC1* mutants could be associated with this role. First, the authors do not present the genome wide transcript level datasets in detail, which they must have obtained for the *ASC1* mutants relative to wild type. Is there any evidence for alterations in mRNA levels correlating with the ribosome densities? Indeed, at one point the authors show in a supplementary figure that translation efficiency correlates with mRNA half-life measurements from the Cramer group. Could this be the reason *ASC1* mutants are altered in their ribosome density on short mRNAs? – Could it be that the role of Asc1 in mRNA decay pathways is the key?

*Reviewer #3:*

In the presented paper, Thompson et al. uncover a role for Asc1/RACK1 in a length-dependent initiation mechanism optimized for efficient translation of genes with important housekeeping functions. It is a compelling story that proposes a simple explanation for what has seemed to be a relatively hardly reconcilable case – wide range of pleiotropic phenotypes associated with deletion of a single gene encoding a ribosomal protein. I also find important that some of many discrepancies/confusions always connected with *ASC1* – like the proposed "ribosome on vs. off" functions of this protein – have been clarified by these authors. But in my opinion one key control that would nail down the highly specific role of *ASC1* in promoting high TE on closed-loop short ORF-containing mRNAs (encoding mainly RPs and MRPs) is missing.

In particular, I tend to disagree with the authors that deletion of *ASC1* per se compromises 60S biogenesis; in fact, this conclusion conversely further broadens the *ASC1* controversy. We, and others, showed that it is exclusively U24 – the *ASC1* intron – that is responsible for maintaining the wt levels of 60S subunits and not the ASC1 protein (Li et al., Plos Biol 2009 and Kouba et al. NAR, 2012). I wonder how the authors measured the 60S/40S ratios? I did not find a single note describing this procedure in the entire paper and the corresponding figure shows only the entire polysome profile that by definition cannot be used for estimating the 60S/40S ratio. This must be monitored in the absence of Mg^2+^ ions and CHX, which I am not sure if it was – visual inspection of free 40S and 60S peaks in Figure 1 seems to suggest that these peaks could have been used for calculations…. Also, M1X cells show no halfmers compared to *asc1∆* or *∆intron* cells (Figure 1—figure supplement 1) further indicating that the 60S biogenesis defect is caused solely by the missing U24. If true, then the entire chapter describing the effects of mutations in two large ribosomal proteins (showing halfmers) seems irrelevant as well as their use in the glycerol media. I would think that the most appropriate way to show that the observed effects are really specific for *ASC1* would be to use mutants of some small ribosomal proteins occurring on the solvent-exposed side of the 40S ribosome in the vicinity of *ASC1* and examine their effects in the same way as shown in Figure 5. Only then the authors could claim that the effect that they observed, which I do not dispute at all, is specific for *ASC1*. What if it is simply caused by a compromised function of any small ribosomal protein that may come in contact with closed-loop mRNAs with short ORFs and/or specific eIFs promoting their recruitment?

---

## [Author Response]

*Your paper has been reviewed by three experts in the field and the general consensus is that it is potentially suitable for publication pending the outcome of additional analyses and revisions. One key issue concerns the view that there is inadequate analysis of protein synthesis from native genes or reporters to back up the claim that translation of short mRNAs with a heightened propensity to form the closed-loop is particularly sensitive to elimination of Asc1. The one reporter you analyzed where the coding sequence length was varied systematically was considered a creative approach, but the results obtained were not very compelling. Suggestions were offered for a different kind of reporter analysis, where a native long mRNA would be systematically shortened, and to examine protein synthesis of a panel of native mRNAs representing Asc1-dependent versus independent genes to confirm the interpretations of the ribosome-profiling data for authentic mRNAs. You could extend the analysis in Figure 4 to include additional Asc1-dependent mRNAs, and also include Asc1 independent mRNAs, but it might be difficult to detect changes in polysome distribution for short mRNAs, in which case measurements of protein and mRNA synthesis or reporter expression would be required. I recall that in our paper on eIF4G depletion (Park et al), which you used creatively in your analysis, we validated a number of mRNAs as being eIF4G-dependent or -independent, and perhaps some of these mRNAs that are short and enriched for both eIF4G and PABP occupancy could be analyzed in your strains to achieve stronger validation of your ribosome profiling data.*

We have improved the quantification of the ORF length reporters by quantifying protein levels from each of three biological replicates with four technical replicates. We now observe a more convincing difference in translational efficiency between the shortest and longest ORFs in the *asc1-M1X* mutant (p = 0.002, Student's t-test) (Figure 3). We saw the same trend as before for the intermediate length reporters, but an unfeasible number of biological and technical replicates would likely be needed to assess the significance of these changes. We therefore excluded the intermediate length reporters from the revised figure. We note that the global relationship between ∆TE and ORF length (Figure 3), predicts a TE reduction for the 600 nt reporter of <15%, which is extraordinarily difficult to measure by quantitative Western blotting.

*Another important issue was whether you have effectively untangled the contributions of mRNA length and propensity to form the closed loop as determinants of Asc1-dependence using bioinformatics and statistical analysis. There was general agreement among the reviewers during the consultation session that you should attempt to bolster your model by showing that an existing eIF4G mutation known to disrupt the closed loop mRNP would dampen the effects of eliminating Asc1 on translational efficiency of Asc1-dependent mRNAs or reporters.*

We agree that determining the effect of Asc1 in a closed loop deficient background is a good experiment, which we attempted using the *eif4g1-N∆300 eif4g2∆* mutant that removes the Pab1 binding site from eIF4G1 (strain obtained from Allan Jacobson, originally described in (Tarun et al., 1997)). Unexpectedly, the *eif4g1-N∆300 eif4g2∆* mutant displayed slightly *increased* rather than decreased translation efficiency of the ‘closed loop’ mRNAs, see Figure 7 (note: for reviewers only). The increase in TE was reduced in the presence of the *asc1∆* mutation, but this result is murky to interpret given the unexpected behavior of the *eif4g1-N∆300 eif4g2∆* mutant by itself. We are concerned that the *eif4g1-N∆300 eif4g2∆* strain, which grows very well in our hands, may have acquired a suppressor that allows robust translation in the absence of the closed loop interaction. We plan to reconstruct this eIF4G mutation in a strain carrying a repressible copy of wildtype eIF4G and/or seek closed loop mutant strains from other sources. We hope that this time-consuming work will eventually be presented elsewhere.

Author response image 1.**DOI:**
http://dx.doi.org/10.7554/eLife.11154.021

*As these eIF4G mutations were made in the S288C background, such experiments would also address a concern that your study employed the Sigma background, which is rarely employed in studies on gene expression. A related point is that, without demonstrating that elimination of at least one other 40S protein in the vicinity of Asc1 on the solvent-exposed surface of the 40S subunit does not have the same effects on TE you observed on elimination of Asc1, it is dangerous to conclude that Asc1 is specifically dedicated to this regulatory function.* We now include ribosome footprint profiling data for two additional mutants affecting small subunit ribosomal proteins in the vicinity of Asc1 near the mRNA exit channel, *rps0b∆* and *rps16b∆* (Figure 5—figure supplement 2) These mutants show growth and bulk translation defects ≥ *asc1* null mutants (Figure 5—figure supplement 2 and data not shown) and cause substantial translational changes globally which do not correlate with the changes observed in *asc1-M1X* (Figure 5). Importantly, the other 40S mutations do not decrease translation of the ‘closed loop’ mRNAs (Figure 5). Note that we do not exclude the hypothesis that other proteins may be involved in regulating the activity of the closed loop; however, our data clearly demonstrate that the effect of Asc1 on ‘closed loop’ mRNAs is not a general phenomenon caused by defects in ribosomal proteins.

*There was also agreement that it is necessary to measure mitochondrial protein synthesis and show that it is reduced in cells lacking Asc1 rather than relying on growth assays in nonfermentable carbon sources.*

We have now assayed mitochondrial protein synthesis in the *asc1-M1X* mutant by 35S incorporation. Synthesis of each mitochondrially-encoded protein decreased by ≥two-fold in the *asc1-M1X* mutant (Figure 6—figure supplement 1). These results are consistent with our model that poor growth in glycerol reflects compromised mitochondrial function downstream of reduced translation of nuclear-encoded mitochondrial proteins in *ASC1* mutants.

*And one of the reviewers is concerned that your conclusion that Asc1 is required for normal 60S biogenesis is at odds with other careful work done previously that ruled out this possibility, and feels that your measurements of 40S:60S subunit ratios from the polysome profiles in Figure 1 do not represent the most rigorous approach to establishing this point. I think it is important to carefully consider these criticisms to avoid "muddying the waters" concerning the involvement of Asc1 in ribosome biogenesis.* Thank you for raising this concern. We have taken care to reword that section of the text to avoid discrediting the previous study (Li et al., 2009). Li et al. concluded that loss of snR24 (also known as U24) was responsible for both the halfmer phenotype and the 60S depletion phenotype of the *asc1∆* mutant because experiments reintroducing intronless *ASC1* and *SNR24* separately showed that *SNR24* expression rescued both the 60S levels and the halfmer phenotype of *asc1∆*. We note that our data are not truly at odds with this finding because our WT Σ1278b polysome profiles have higher free 60S subunit levels compared to 40S subunits. The fact that loss of the Asc1 protein reduces the free 60S/40S ratio may be specific to the Σ1278b background, and we did observe that loss of *SNR24* was responsible for the halfmer phenotype of the *asc1∆* mutant at 37 ˚C, in agreement with the previous study.

*Another of the reviewers was justifiably concerned that you may have overlooked the effects of deleting Asc1 on "no-go" decay and that defects in this pathway and attendant changes in progression of stalled ribosomes on no-go substrates, or changes in mRNA levels, could be influencing conclusions about alterations of translational efficiency. It is important to consider this possibility and more fully document the effects of eliminating Asc1 on mRNA levels and the extent to which changes in TE might be driven by defects in no-go decay.* The observed changes in gene expression and translational efficiency are not consistent with defects in no-go decay. If decreases in translation efficiency were caused by defects in no-go decay stabilizing mRNAs (thus inflating the denominator in the footprint RNA/total RNA calculation), then we would expect the levels of those mRNAs to increase in the total RNA pool. However, the overall trend is for the levels of total mRNA for mRNAs with decreased TE in the *asc1-M1X* mutant to go down or remain constant rather than increase (Figure 4, Figure 4—figure supplement 2). We have added a short discussion of this effect in the text as well as a supplemental figure (Figure 4—figure supplement 2).

*Finally, there were some issues with a lack of adequate descriptions of procedures and bioinformatics analysis.* Thank you for bringing this to our attention. We have expanded our descriptions of procedures and analyses to address these concerns (see detailed descriptions below in the point-by-point response).

Reviewer #1:

*General critique:*

*It is quite interesting that Asc1 is found to be particularly important for efficient translation of short mRNAs and also mRNAs with strong potential for closed-loop formation, and it is noteworthy that they have discovered in previously published datasets unsuspected relationships between short mRNA length and both high closed-loop forming potential and heightened dependence on eIF4G. However, since short mRNAs are enriched for both closed-loop formation potential and for dependence on eIF4G, it isn't entirely clear whether it is length per se, ability to form the closed loop, or dependence on eIF4G that confers the greater than average requirement for Asc1 for efficient translation.* We were also quite struck by the relationship between ORF length and closed loop formation and its potential to explain the high translational efficiency of short ORFs observed in ribosome profiling data from diverse eukaryotes. We agree that it is difficult to say using bioinformatics alone what the direction of causality is between Asc1, eIF4G, and the closed loop. Indeed, if they all act on the same mRNAs as part of a common pathway, it will be difficult to assign specific effects to each individual component. We are interested in this problem, but believe that it will take years of careful experiments to fully disentangle the effects of each player in the translational regulation of ‘closed loop’ mRNAs. However, we do show that the decrease in TE of the closed loop genes in the *asc1-M1X* mutant cannot be accounted for by a simple linear relationship between ∆TE and ORF length (Figure 3), and we have explained this analysis more thoroughly in the Methods section under the heading “ORF length correction of ∆TE values for closed loop groups”.

*Specific comments: The key experiment in Figure 3 should be bolstered with additional replicates in an effort to increase the statistical significance of differences between the monomer vs dimer or trimer.*

*If that can't be accomplished, then an additional, complementary experiment could be conducted where they start with a native, long mRNA that is Asc1-independent and show that Asc1-dependence is conferred by progressive removal of coding sequences. Perhaps this could be done for an mRNA whose protein product is a scaffold comprised of multiple modular domains.*

We have included additional replicates of the ORF length reporter experiments and now show more significant differences (p=0.002). Please see the response to this point in the summary review above.

*An important experiment that would test their conclusion that Asc1 is important for enhanced translation of mRNAs in the closed-loop configuration would be to examine the effect of deleting Asc1 in a previously published strain in which the PABP binding site in eIF4G1 is deleted, eliminating its ability to form the closed -loop intermediate, and eIF4G2 is deleted. (Tarun and Sachs showed in 1997 that the entire N-terminus of eIF4G1 can be deleted in a strain lacking eIF4G2 with only moderate effects on cell growth.) Their model strongly predicts that the effect of deleting Asc1 on translation of short, "strong closed-loop mRNAs" will be substantially dampened in this eIF4G1 mutant.*

We attempted this experiment, with suggestive but ambiguous results potentially due to problems with the *eif4g1-N∆300 eif4g2∆* mutant strain we obtained. Please see our detailed response in the summary above.

*They do not show directly that mitochondrial protein synthesis is impaired in asc1 cells and cannot rule out effects on expression of other genes required for growth in nonfermentable carbon sources. It seems necessary to measure mitochondrial protein synthesis directly in asc1 cells, which is a feasible experiment, to justify their claims.* We now provide these data. Please see our response in the summary above.

Reviewer #2:

*In this manuscript the authors assess the role of the ASC1/RACK protein in protein synthesis. They use the ribosome footprinting technology to address the impact of different mutants in ASC1 and conclude that these mutations preferentially alter the translation of short mRNAs. I found the data interesting but oversold in places. The paper was also quite difficult to read due to a lack of technical detail in many of the methods descriptions and legends. Finally, I was not convinced that the authors had effectively shown that Asc1 affected translation directly. The authors desperately need to provide some direct measure of protein synthesis in this paper. The ribosome footprinting pattern across mRNAs provides an approximation of translation and how this changes, but there are other reasons that ribosomes might accumulate on mRNAs (see point about no-go mRNA decay below). So claims are made about the rates of initiation and the role of Asc1 in protein synthesis without actually ever measuring the synthesis rate of any endogenous proteins directly. In Figure 3, the authors do measure the translation efficiency of a series of artificial engineered proteins to study the correlation between ORF length and the impact of ASC1 mutation. However, they need to show the primary data here so that the impact of the mutation on protein synthesis can be directly assessed by the reader. Also in my view validation that Asc1 specifically affects the synthesis of protein from some short mRNAs relative to some longer ones is essential.* We have validated the effect of Asc1 on the translational efficiency (protein/mRNA) of a short vs. long ORF reporter, and now include many more replicates demonstrating the significance of the length-dependent difference in sensitivity to Asc1. The primary data for the ORF length reporter experiments is quantitative Western blotting using fluorescent antibodies and qRT-PCR. A representative Western blot is shown in Figure 3—figure supplement 3 to show the reader the clean nature of the fluorescent signal used to determine protein abundance. Since it is difficult to quantify protein abundance changes on this scale by eye, presentation of all the Western blots does not seem likely to be very helpful to the reader.

We are not sure what the reviewer had in mind for a “direct measure of protein synthesis” for a specific gene.

*The choice of strain for this study is very strange. While the Σ1278b strain is widely used for studies into filamentous growth in S. cerevisiae it is quite rarely used outside this, and it has some odd characteristics that raise concerns over the generality of the results shown. As has been shown previously on a number of occasions and is now shown in the polysomes presented here, Σ1278b is very unusual in that it has abnormally high levels of free 60S subunits. The asc1 null has previously been shown to decrease 60S subunit levels and again the authors reconfirm this observation. The authors state that the ASC1 mutants have relatively subtle effects on protein synthesis and make statements about how subtle effects on translation as judged by polysome profiling can conceal a wider perturbation of cellular translation. It is possible that the abnormal ratio of free 40S/60S subunits in the Σ strain obviates potentially more dramatic effects of the ASC1 mutation on translation. In my view the authors should really test key observations from their study in a laboratory strain of yeast that has a more standard profile of free 40S and 60S subunits. Obviously this may be an unrealistic expectation as ribosome footprinting is neither cheap nor trivial- but some attempt to generalize the data could be made and the non-standard nature of the strain and the ratio of ribosomal subunits should be commented upon.* Our laboratory has done several previous studies using the Σ1278b strain, which, because it is wildtype for *FLO8*, retains characteristics of natural isolates that make it attractive for study. The *asc1∆* mutant has a striking phenotype, complete loss of invasive growth, in the Σ1278b background. We therefore began our studies in this strain. There seems to be no firm basis for characterizing the free 40S/60S ratios of Σ1278b, S288c, or w303 as ‘normal’ for yeast, although it is interesting that they differ. We are ultimately interested in looking at the conserved function of Asc1 in other eukaryotic species rather than other strains of yeast. The conservation of Asc1 and the closed loop factors suggests that this function of Asc1 should be highly conserved.

*For the polysomes presented in Figure 1 the polysomes profiles have very high 80S peaks relative to the polysome peaks for unstressed cells, this suggests that some stress has occurred during the polysome preparation? Could this be masking the impact of the asc1 mutations on polysome association?* These polysome profiles are typical for unstressed profiles from our lab. However, even if there has been some stress during polysome preparation, both the wildtype and the mutant cultures were processed identically.

*Figure 2 – I find this formaldehyde polysome analysis particularly unconvincing. The polysomes are barely visible and the pattern of the other peaks looks substantially altered. The authors claim that Asc1 is maintained across the gradient in the mutants seems inappropriate based on the data presented and based on what is in the literature.* We used previously published and widely accepted methods for formaldehyde crosslinking of polysomes (see (Valásek et al., 2007)), and our crosslinked polysomes look similar to those described in this reference. We hypothesize that the polysomes are lower because some fraction of them are crosslinked to larger complexes that do not resolve in the sucrose gradient. We agree that crosslinking is not quantitative and only seek to make the qualitative claim here that the data support some remaining association of the *ASC1* mutants with the ribosome in vivo; thus it cannot be assumed that these mutations fully abrogate ribosome binding.

*As the authors state, Asc1 has also been shown by the Inada group to play a role in the no-go mRNA decay process. It promotes translation arrest to facilitate mRNA decay by this pathway. It is therefore possible that the alterations in ribosome density on mRNAs for ASC1 mutants could be associated with this role. First, the authors do not present the genome wide transcript level datasets in detail, which they must have obtained for the ASC1 mutants relative to wild type. Is there any evidence for alterations in mRNA levels correlating with the ribosome densities? Indeed, at one point the authors show in a supplementary figure that translation efficiency correlates with mRNA half-life measurements from the Cramer group. Could this be the reason ASC1 mutants are altered in their ribosome density on short mRNAs? – Could it be that the role of Asc1 in mRNA decay pathways is the key?* Defects in no-go decay are inconsistent with the effects we see on ∆TE. Please see our complete response in the summary above. We bring up the half-life correlations because we speculate that mRNA decay downstream of translational repression could explain the decrease in total RNA levels for some mRNAs with decreased TE in the *ASC1* mutants. A full understanding of this effect is outside the scope of this study, but recognizing the effect is an important aid to future discussion.

Reviewer #3:

*In the presented paper, Thompson et al. uncover a role for Asc1/RACK1 in a length-dependent initiation mechanism optimized for efficient translation of genes with important housekeeping functions. It is a compelling story that proposes a simple explanation for what has seemed to be a relatively hardly reconcilable case – wide range of pleiotropic phenotypes associated with deletion of a single gene encoding a ribosomal protein. I also find important that some of many discrepancies/confusions always connected with ASC1 – like the proposed "ribosome on vs. off" functions of this protein – have been clarified by these authors.*

We are glad this reviewer finds our story compelling. We agree that clarifying the behavior of Asc1 mutant proteins, which are often interpreted as being completely “off” the ribosome, will be important for progress in the field.

*But in my opinion one key control that would nail down the highly specific role of ASC1 in promoting high TE on closed-loop short ORF-containing mRNAs (encoding mainly RPs and MRPs) is missing. In particular, I tend to disagree with the authors that deletion of ASC1 per se compromises 60S biogenesis; in fact, this conclusion conversely further broadens the ASC1 controversy. We and others showed that it is exclusively U24 – the ASC1 intron – that is responsible for maintaining the wt levels of 60S subunits and not the ASC1 protein (Li et al., Plos Biol 2009 and Kouba et al. NAR, 2012). I wonder how the authors measured the 60S/40S ratios? I did not find a single note describing this procedure in the entire paper and the corresponding figure shows only the entire polysome profile that by definition cannot be used for estimating the 60S/40S ratio. This must be monitored in the absence of Mg^2+^ ions and CHX, which I am not sure if it was – visual inspection of free 40S and 60S peaks in Figure 1 seems to suggest that these peaks could have been used for calculations…. Also, MX1 cells show no halfmers compared to asc1∆ or ∆intron cells (Figure 1—figure supplement 1) further indicating that the 60S biogenesis defect is caused solely by the missing U24. If true, then the entire chapter describing the effects of mutations in two large ribosomal proteins (showing halfmers) seems irrelevant as well as their use in the glycerol media.*

Please see our comments above regarding our interpretation of the halfmer phenotype and 60S subunit levels in the *asc1∆* and *asc1-M1X* mutants and the likelihood of strain background effects in the differences between studies. We have now clarified the method used for 60S/40S ratio quantification in the Methods section. You are right in that we quantified the native polysome profiles directly without subunit dissociation. We did so because pilot experiments using subunit dissociation did not appear to be as sensitive to small changes as using the whole polysome profile. It is expected that a slight subunit imbalance will be more apparent in the whole polysome profile because only unassociated 40S and 60S subunits will be quantified. Although we cannot interpret the obtained values at the literal 60S/40S ratio (as in the case of polysome dissociation), we see no reason that they cannot be compared between different mutants as a way to assess their subunit imbalance.

*I would think that the most appropriate way to show that the observed effects are really specific for ASC1 would be to use mutants of some small ribosomal proteins occurring on the solvent-exposed side of the 40S ribosome in the vicinity of ASC1 and examine their effects in the same way as shown in Figure 5. Only then the authors could claim that the effect that they observed, which I do not dispute at all, is specific for ASC1. What if it is simply caused by a compromised function of any small ribosomal protein that may come in contact with closed-loop mRNAs with short ORFs and/or specific eIFs promoting their recruitment?*

We agree that it was important to fully demonstrate that Asc1 has a specific function in promoting translation of the ‘closed loop’ mRNAs. We have now characterized two additional small ribosomal subunit protein mutants in the vicinity of Asc1 as suggested by this reviewer. We found that they do not resemble the *asc1-M1X* mutant in any of the important particulars. Please see our discussion of these results in the summary above.